# Compact-Fusion Feature Framework for Ethnicity Classification

**Tjokorda Agung Budi Wirayuda [1,2,\*], Rinaldi Munir [3,\*] and Achmad Imam Kistijantoro [3]**

[1]  Doctoral Program of School of Electrical Engineering and Informatics, Bandung Institute of Technology, Bandung 40257, Indonesia
[2]  School of Computing, Telkom University, Bandung 40227, Indonesia
[3]  School of Electrical Engineering and Informatics, Bandung Institute of Technology, Bandung 40166, Indonesia; imam@staff.stei.itb.ac.id
[\*]  Correspondence: cokagung@telkomuniversity.ac.id (T.A.B.W.); rinaldi@informatika.org (R.M.)

**Abstract:** In computer vision, ethnicity classification tasks utilize images containing human faces to extract ethnicity labels. Ethnicity is one of the soft biometric feature categories useful in data analysis for commercial, public, and health sectors. Ethnicity classification begins with face detection as a preprocessing process to determine a human's presence; then, the feature representation is extracted from the isolated facial image to predict the ethnicity class. This study utilized four handcrafted features (multi-local binary pattern (MLBP), histogram of gradient (HOG), color histogram, and speeded-up-robust-features-based (SURF-based)) as the basis for the generation of a compact-fusion feature. The compact-fusion framework involves optimal feature selection, compact feature extraction, and compact-fusion feature representation. The final feature representation was trained and tested with the SVM One Versus All classifier for ethnicity classification. When it was evaluated in two large datasets, UTKFace and Fair Face, the proposed framework achieved accuracy levels of 89.14%, 82.19%, and 73.87%, respectively, for the UTKFace dataset with four or five classes and the Fair Face dataset with four classes. Furthermore, the compact-fusion feature with a small number of features at 4790, constructed based on conventional handcrafted features, achieved competitive results compared with state-of-the-art methods using a deep-learning-based approach.

**Keywords:** compact-fusion feature framework; conventional handcrafted features; ethnicity classification; feature fusion; optimum feature

## 1. Introduction

Today, the development of technology for data collection, especially image data, provides an environment that supports demographic data extraction from face images. The demographic data in this paper focus on ethnicity classification, in which, to label ethnicity, the ground truth uses labels provided from a public dataset to calculate the classification accuracy. Compared to other demographic data, ethnicity classification has been studied the least [1,2]. Nevertheless, it has received increasing attention in recent years, primarily due to its numerous applications, such as in biometrics identification, video surveillance, forensic art, human–computer interaction, targeted advertisements, social media profiling, extensive database searching, demographic statistics, and social science research (understanding human social behaviors and their relations to the demographic backgrounds of individuals) [1–3].

Different approaches have been proposed for ethnicity classification, focusing on feature representation and accuracy for ethnicity classification, especially on deep-learning architecture [2]. The best state-of-the-art result (SOTA) for ethnicity classification is achieved using solutions based on convolutional neural networks (CNNs) [1,2,4,5]; however, the disadvantage of this deep-learning approach is that it requires high-cost computation resources. Meanwhile, conventional solutions based on handcrafted features have been shown to provide comparable accuracy with low-cost computing demand. For example,

the study by Becerra-Riera et al. [1] reported that handcrafted features achieved comparable accuracy to the CNN approach for ethnicity classification in the FERET subset from the EGA database. A promising alternative was reported in [1] for the development of ethnicity classification based on the handcrafted-features approach; however, the comparisons were not fair enough because the dataset was relatively small and was insufficient for the deep-learning approach to function optimally. The gap between handcrafted and deep-learning solutions was shown in the study by Al-Azani and El-Alfy [6] using the HOG feature for ethnicity classification in a large dataset, which achieved an accuracy level of 69.86% for three ethnicity classes; this lagged far behind the deep-learning approach, which reached an accuracy level of over 80.00% [2]. The proposed method, which is based on handcrafted features, addresses this gap by testing ethnicity classification in a large dataset, especially those collected in unconstrained environments with three or more ethnicity classes.

The main goal of ethnicity classification is to accurately determine an ethnicity label from unseen face images. The handcrafted-feature approach is used with the aim of generating high-dimensional features, commonly by combining one or two handcrafted-feature types. Combining one or two handcrafted features provides a more discriminant feature to achieve higher accuracy [1,6]. Feature reduction is applied to combined features to produce small-size features, which retain the advantages of the handcrafted-features approach by maintaining a low-cost resource demand. However, an extreme feature-reduction process causes a significant decrease in accuracy and eliminates the semantic meaning of handcrafted features. This paper provides an overview of recent feature representation focused on feature-reduction methods (selection and learning) and proposes a framework to produce the compact-fusion feature for ethnicity tested on a large dataset. The contributions of this paper are as follows:

(i)    A handcrafted-feature solution for ethnicity classification that was tested on two large datasets with five ethnicity classes (White, Black, Asian, Indian, and Others) and four ethnicity classes (White, Black, Asian, and Indian) is implemented;

(ii)   The proposed framework utilizes four handcrafted features (multi-local binary pattern (MLBP), histogram of gradient (HOG), color histogram, and speeded-up-robust-features-based (SURF-based)) as the basis for generating a compact-fusion feature with highly discriminant information;

(iii)  The proposed framework performs feature reduction only using a single data instance with a simple selection strategy, providing a better understanding of data representation and generation.

Tested on two large datasets with five ethnicity classes (White, Black, Asian, Indian, and Others) and four ethnicity classes (White, Black, Asian, and Indian), the proposed framework achieves high accuracy with a minimum feature size compared with other methods.

Section 2 reviews feature representation through feature reduction and ethnicity classification based on handcrafted features and discusses the dataset problem and standard feature-extraction method used. Section 3 describes the proposed framework focused on data transformation from raw data to final feature representation, feature extraction, and the detailed process at each stage. Section 4 describes the datasets used and detailed results of exhaustive experiments. Finally, the last section attempts to draw conclusions based on the results of the conducted experiment and future research directions.

## 2. Related Works

This section discusses related works regarding ethnicity classification based on handcrafted features and compact feature representation, which provided the foundation for the development of the proposed framework.

### 2.1. Ethnicity Classification Based on Handcrafted Features

This section contains two sections of a focused literature review. The first section discusses research related to ethnicity recognition based on handcrafted features and focuses

on early papers that performed ethnicity recognition on large datasets. The second focuses on research related to the single/multi-feature and fusion strategy. Both sections discuss and analyze the research with regard to the number of classes, number of data, ethnicity label distribution, feature-extraction method, feature reduction, and performance measurement. Finally, a summary of the selected papers that discussed ethnicity classification based on handcrafted features is shown in Table 1.

**Table 1.** Summary of ethnicity classification based on handcrafted features.

| Paper | Year | Handcrafted Feature-Extraction Method | Ethnicity | Feature Reduction | Classifier | Dataset | Accuracy (in %) |
|---|---|---|---|---|---|---|---|
| [7] | 2010 | BIF (Gabor + Maximum Value) | Asian, Black, Hispanic, White, and Indian | PCA + OLPP | Manifold Learning | MORPH-II with 55,127 images | Mean class 72.73 |
| [8] | 2015 | Single Feature: LBP, CLBP, HOG, SWLD | White, Black, Asian, and Hispanic | LDA | SVM | Merging MORPH-II with FERET resulted in 55,195 images | 79.40 using CLBP |
| [9] | 2019 | SIFT + Fisher Vector | Asian, Black, Hispanic, and White | PCA | DAG-TIPTAC | MORPH-II with 55,068 images | Mean class 89.61 |
| [10] | 2020 | Single Feature: SURF, LBP, HOG, Color Moment | Asian, Black, Hispanic, White, and Indian | PSO | SVM | MORPH-II with 55,127 images | 93.17 using LBP |
| [11] | 2017 | Fusion from LBP, HOG | Middle Eastern and Non-Middle Eastern | - | Best using SVM | Part of FERET with 2790 images | 98.5 |
| [1] | 2019 | Filterbank, geometrical features, and Combined Feature | African American, Asian, Caucasian, Indian, and Latin | PCA | SVM + Random Forest | EGA dataset with 2345 images | 87.00 |
| [12] | 2021 | Gray Level Co-occurrence Matric, Color Histogram, and Combined Feature | Banjar, Bugis, Javanese, Malay, and Sundanese | - | Random Forest | Private with 2290 images | 98.65 |
| [6] | 2019 | HOG | Asian, Indian, and Others | - | SVM | UTKFace Part 1 with 4109 images | 69.68 |
| [13] | 2023 | Skin Color Palette | Caucasian and Indian | - | SVM | P-DESTRE dataset. Video tracking | 98.00 |

From studies in the literature, the study by Guo and Mu [7] describes the first ethnicity classification that was performed on a very large dataset consisting of 55,000 images, namely, the MORPH-II dataset. The ethnicity classification was performed on five classes, namely, Asian, Black, Hispanic, White, and Indian. The feature-extraction approach used was biologically inspired feature (BIF), which is based on Gabor filters with 4 orientations and 16 scales, which are then down-sampled by taking the maximum values within a local spatial neighborhood and across the scales within a band (two consecutive scales). The Gabor filter-based features result in high-dimensional features; therefore, in Guo and Mu's study, feature reduction using principal component analysis (PCA) was carried out, followed by orthogonal locality preserving projections (OLPP) to produce smaller-sized features. It was reported that the testing resulted in correct estimation levels for Black, White, Hispanic, Asian, and Indian ethnicities of 98.30%, 97.10%, 74.20%, 59.50%, and 6.90%, respectively. The mean class accuracy that we calculated based on data in the paper was 72.73%.

Carcagnì et al. [8] also used MORPH-II for ethnicity classification with additional data from the FERET dataset. The authors investigated the use of single features, including LBP, compound local binary pattern (CLBP), HOG, and spatial Weber local descriptor (SWLD),

with linear discriminant analysis (LDA) used as the dimension reduction method. SVM was used as the classifier, and four scenarios were tested: (1) unbalanced data (original data) without LDA scaling, (2) unbalanced data (original data) with LDA scaling in the range of [0, 1], (3) balanced data (each ethnicity has the same number of samples, i.e., 600) without LDA scaling, and (4) balanced data (each ethnicity has the same number of samples, i.e., 600) with LDA scaling in the range of [0, 1]. The reported average accuracy levels for the recognition of five ethnicities in the four scenarios using LBP, HOG, SWLD, and CLBP were 71.50%, 56.10%, 62.60%, and 72.40%, respectively. Interestingly, the level of accuracy in the balanced data scenarios decreased significantly compared to the scenarios using unbalanced data due to the drastic reduction in the sample size for each ethnicity, which could not represent the variances in ethnicity, age, and gender in the dataset used.

We analyzed subsequent research on the MOPRH-II dataset carried out in [9], which used four ethnicities: Asian, Black, Hispanic, and White. The author explored a single-feature approach that applied Fisher vector encoding to the compressed and augmented scale invariant feature transform (SIFT). The author applied PCA to reduce the high-dimensional feature and used a decision direct acyclic graph-truncated isotropic principal component analysis classifier (DAG-TIPTAC) as the classifier. The author reported the mean correct estimation value of 89.61%, which was higher than that reported by Guo and Mu [7], which was 82.3%. The last paper we analyzed that used the MORPH-II dataset was [10]. The author explored single features, including SURF, LBP, HOG, and Color Moment, with particle swarm optimization (PSO) as the feature-selection method. The highest level of accuracy using SVM as the classifier was reported to be 93.17%, 92.34%, 81.48%, and 80.68% for LBP, Color Moment, SURF, and HOG.

The second review of related research focuses on the multi-feature and fusion-at-feature-level strategy for ethnicity classification. In the fusion-at-feature level, several features are concatenated to produce high-dimensional features, for example, LBP and histogram of gradient (HOG) [11]; filterbank and geometrical features [1]; and gray level co-occurrence matric and color histogram [12]. The study by Mohammad and Al-Ani [11] suggested the fusion feature of LBP with the histogram of gradient (HOG) in the ocular face area for ethnicity classification on the FERET database with 2730 images from 989 subjects using two unbalanced classes (non-Middle East, 94%; and Middle East, 6%). It reported that combining these features would increase the detection power, resulting in higher accuracy. Moreover, four classifiers, support vector machine (SVM), multi-layer perceptron (MLP), linear discriminant analysis (LDA), and quadratic discriminant analysis (QDA), were already tested, resulting in the best result with an overall test performance accuracy level of 98.5% using SVM with the polynomial kernel. However, although a high level of accuracy was achieved, considering the number of classes, the small-sized dataset, and the highly unbalanced class, this method still needs to be extensively investigated before it is applied in real-world scenarios.

Meanwhile, a study by Becerra-Riera et al. [1] observed the combination of filter-bank and geometrical features as feature representations with random forest (RF) and support vector machine as the classifiers. The study used the EGA dataset, which contains 2345 images that have five ethnicity labels. The highest level of accuracy, in general, was 87% using a late fusion strategy from RF with filterbank and SVM with the geometric feature. The results showed only 0.3% less accuracy than the CNN approach, mainly due to the balanced class, and there were no illumination problems in the dataset. Recently, a study by Putriany et al. [12] applied a combination of gray level co-occurrence matric, color histogram, and random forest for ethnicity classification in the five largest ethnic groups in Indonesia: Banjar, Bugis, Javanese, Malay, and Sundanese. They collected a balanced dataset of an average of 458 images for each ethnic group, with a total of 2290 images, and reported that the proposed method achieved 98.65% accuracy. Unfortunately, no comparison analysis was provided because the study used a private dataset.

The study by Al-Azani and El-Alfy [6] developed ethnicity classification by combining the HOG feature with SVM Classifier, which was tested on UTKFace Part-1 with

4109 images labeled in three classes. The images were normalized into $68 \times 68$ pixels; then, the HOG feature was extracted from the red–green–blue color channel (RGB) using nine orientation bins with $8 \times 8$ cells. The final feature vector size was estimated to be 5292, which was extracted using the aforementioned configuration. Using the HOG feature with the SVM classifier, the level of accuracy was 69.86% for three ethnicity classes.

According to the first section of the review related to ethnicity classification and datasets, it can be concluded that there is no consensus regarding the number of classes and minimum data that should be used, even using the same dataset. The maximum number of classes is five when highly unbalanced data are used. This paper uses four and five classes in a large dataset with appropriate balance distribution to evaluate the proposed feature-representation method for ethnicity classification.

From the second section of the review, it can be seen that previous studies have reported high levels of accuracy in ethnicity recognition by combining high-dimensional features, feature reduction, and feature fusion. In [8,10], the single-feature approach based on the LBP feature achieved higher levels of accuracy than the other feature types. Meanwhile, the fusion-at-feature-level approach commonly only uses two types of features [1,7,9,11,12]. The proposed framework uses four types of handcrafted features that represent textures (MLBP and HOG), color distribution (color histogram), and pixel/point patterns (SURF) to provide better discriminant information. Meanwhile, PCA became a popular method used to perform feature reduction due to its robustness and flexibility in controlling reduction size. Finally, all the papers reported classification performance with accuracy.

### 2.2. Compact Feature Representation

A study by Xie et al. [13] categorized feature selection and feature-learning methods as the primary process to produce compact features; compact features were addressed as the result of the feature-reduction process. The feature-selection method aims to reduce the dimension by selecting the subset of features with highly discriminant information. Meanwhile, the feature-learning method applies a transformation to generate a set of new features from the original features; learned features represent the original data and possess suitable properties, such as extracted hidden patterns [13]. The main benefits of feature selection are that it avoids the pitfalls of dimensionality to improve prediction performance, provides a faster and more cost-effective process, and provides a better understanding of data generation through visualization [13].

This paper focuses on a compact-fusion feature framework using a feature-selection approach for the classification task, with ethnicity classification becoming the selected domain for implementation. Summaries of the selected papers that discuss compact feature representation for classification tasks are shown in Table 2.

**Table 2.** Summary of compact feature representation literature.

| Paper | Year | Task Domain | Compact Feature Strategy/Method |
|-------|------|-------------|---------------------------------|
| [14] | 2015 | Age estimation | PCA on the combined features of MLBP and Gabor filter. |
| [15] | 2016 | Image recognition and retrieval | Supervised mutual information based on entropy. |
| [16] | 2018 | Texture classification | Select local binary pattern (LBP) histograms or bins. |
| [17] | 2021 | Face classification | The maximum magnitude selection from HOG. |
| [18] | 2022 | Age estimation | The maximum value from the Gabor filter response. |
| [19] | 2022 | Face recognition | Select two main edge directions with the highest magnitudes. |
| [20] | 2022 | Gender recognition | DWT from mean Gabor response. |

A study by Nguyen et al. [14] proposed an age-estimation approach by categorizing images into five groups based on their blurring degree and generating a learning model for each group by combining multi-local binary pattern (MLBP), Gabor filter, prin-

cipal component analysis (PCA), and support vector regression. PCA serves as a feature-reduction method on the combined features of MLBP and the Gabor filter. The study by Zhang et al. [15] proposed a compact feature for image classification tasks selected using supervised mutual information based on entropy; it was applied to the Fisher vector (FV) and vector of locally aggregated descriptors (VLAD). The study reported that the proposed method achieved a higher level of accuracy and required a lower computational cost than feature compression methods such as product quantization. PCA and mutual information, based on entropy, process the features from all training data for the transformation/selection process.

The study by Porebski et al. [16] proposed a multi-color space histogram selection (MCSHS) and a multi-color space bin selection for texture classification. These approaches select local binary pattern (LBP) histograms or bins processed from images coded in multiple color spaces. In another study by Nuyen-Quoc and Hoang [17], the information was reduced by selecting a binned histogram of gradient (HOG) using the maximum magnitude selection method for the face classification task, and this was performed in a single-image feature vector. The maximum value process was also used in the study by Lu et al. [18] to fuse the Gabor feature at five scales and eight directions that contained redundancy in the feature data. The maximum value resulted in 5 compact two-dimensional features from the initial 40 two-dimensional features, and the process was conducted using a single image. Furthermore, the study by Najmabadi and Moallem [19] proposed feature selection by selecting two main edge directions with the highest magnitudes as compact features. As a result, the proposed method achieved the highest recognition rate for face recognition tasks on experiment data. The study by Gupta et al. [20] proposed a mean DWT feature optimization method for large-dimension Gabor filters to produce compact feature vectors with minimal redundancy. Combined with SVM, the proposed compact and minimal-redundancy feature vectors achieved a high level of accuracy of 99.5% in a gender classification task on the UTKFace child dataset.

According to this review related to compact feature representation, it can be concluded that compact features from high-dimensional features tend to minimize redundant information by generating a new set of features through the transformation learned from all data or selecting features based on feature ranking or highest scores. From the scope of data needed to create a compact representation, the methods can be categorized into whole data methods [14–16] (which need to provide a training set for the process) and single data methods (which only need feature vectors from a single image) [17–20]. The proposed framework produces compact feature representation using a single data instance approach and applies feature selection based on the maximum and average values from rearranged feature vectors.

## 3. Proposed Framework

### 3.1. Overview of General Process

In general, the solution for ethnicity classification consists of three major stages: a dataset preprocessing phase, a training phase, and a testing phase. The proposed framework became a training part of the general process, as seen in Figure 1.

Here, the dataset preprocessing process consists of face detection and image normalization. Face detection is applied in the dataset preprocessing phase to ensure that each image from the dataset contains a human face. Then, after confirming the presence of a face in the images, they are normalized into $200 \times 200$ pixel sizes, followed by an adaptive histogram equalization process, and the images are saved as a preprocessing dataset. Meanwhile, the image data that fail to be detected as a front-facing human face are excluded from the preprocessing dataset. Finally, the preprocessing dataset is split into training, validation, and testing sets to develop and evaluate the proposed method.

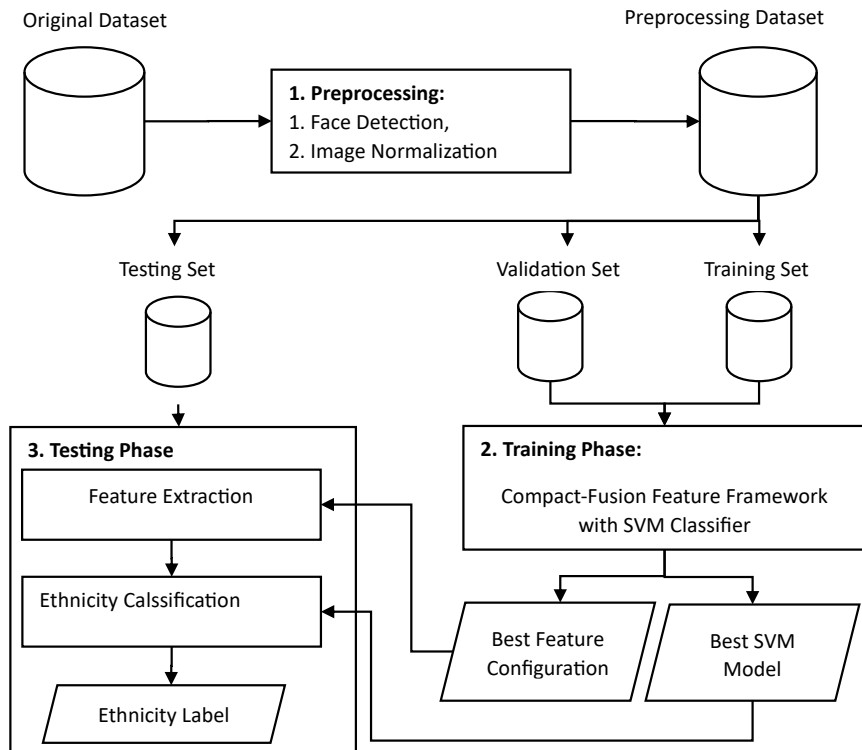

**Figure 1.** General process used to develop the proposed framework.

As part of the training phase, the proposed framework consists of three main stages: independent parameter tuning, compact feature strategy, and feature fusion, as seen in Figure 2. The proposed method is configured and evaluated using training and validation sets during the training phases. The training and evaluation processes are carried out at each stage to obtain the best feature configuration as the input for the next step. The final results from the training phase are the SVM model ethnicity classifier and best feature configuration to extract compact-fusion features.

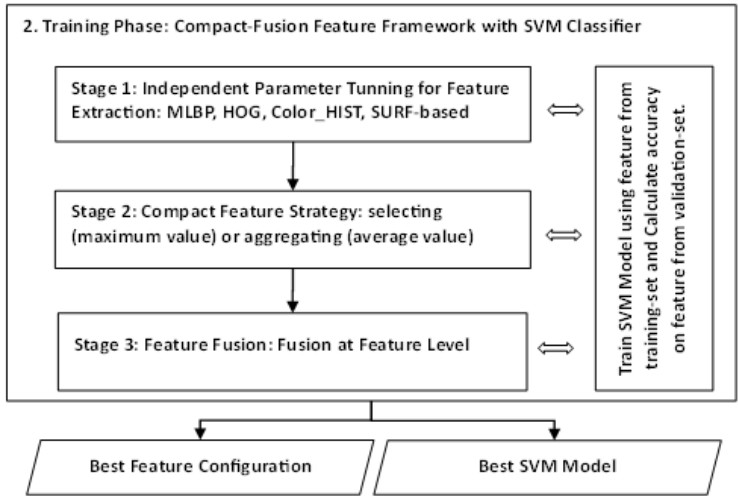

**Figure 2.** Proposed framework: compact-fusion feature.

In the testing phase, the compact-fusion feature framework is applied to the testing set and classified using the SVM model from the training phase. Then, several exhaustive experiments are conducted on the testing set to comprehensively evaluate and analyze the proposed framework.

In short, as seen in Figure 3, the proposed framework transforms raw data into feature representation through independent parameter tuning (optimal feature selection), compact feature strategy/extraction, and fusion at the feature level (compact-fusion feature representation). First, the high-dimensional handcrafted features are extracted from the image, and then, the optimal feature selection produces the optimum features from the high-dimensional handcrafted features, which balance size and accuracy. The next stage is to produce compact features with low dimensions and acceptable accuracy levels. Finally, each handcrafted feature is concatenated to produce a compact-fusion feature with a high accuracy level and minimum feature size. The handcrafted-feature methods used in the proposed framework are explained in Section 3.2. The detailed process from independent parameter tuning, compact feature strategy, and fusion at the feature level is described in Sections 3.3–3.5, respectively.

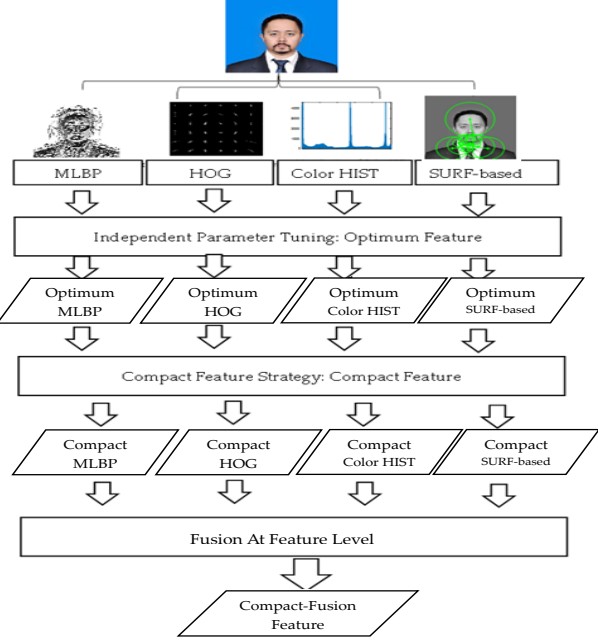

**Figure 3.** Illustration of feature transformation in the proposed framework.

Written informed consent was obtained from the study participants for the publication of their details.

The parameters involved in the proposed method were the grid-size value for each feature-extraction method and the selection strategy used for compact features (the maximum selecting method and the average value method). The advantages of the proposed compact-fusion feature framework compared to existing feature-reduction approaches are as follows:

1.  The high-dimensional feature is generated from parameter variation in handcrafted features' extraction, resulting in an effective compact feature process because the feature vector is in the same domain;
2.  The selection strategy for compact features only utilizes information from single-row feature vectors without involving other rows. It is performed with a simple selection strategy, resulting in a compact feature with a better understanding of data representation and generation;
3.  The compact feature process is applied before the fusion feature, resulting in a compact feature with the optimum performance used for the fusion;
4.  The multiple features used represent textures, colors, and pixel patterns that enrich feature vectors for better discriminant capability.

*3.2. Handcrafted-Feature Methods*

The proposed framework uses four handcrafted feature-extraction methods selected based on the literature review, consisting of a multi-local binary pattern (MLBP) [8,11,14], histogram of gradient (HOG) [6,8,11], color histogram (Color HIST) [12,21], and speeded-up-robust-features-based (SURF-based) [10].

3.2.1. Multi-Local Binary Pattern (MLBP)

The multi-local binary pattern is a variation of the local binary pattern (LBP) that provides better discriminant information by combining several LBP features obtained through variations in radius, the number of neighbors, and the number of sub-blocks [14]. The multi-local binary pattern (MLBP) consists of several single-level LBP features with different parameters of radius (R), number of surrounding pixels (P), and number of sub-blocks (M) applied to the image. Therefore, by combining several LBP features obtained through variations in radius, the number of neighbors, and the number of sub-blocks, the information obtained will be richer and able to capture global and local characteristics. Figure 4 illustrates the multi-local binary pattern (MLBP) process.

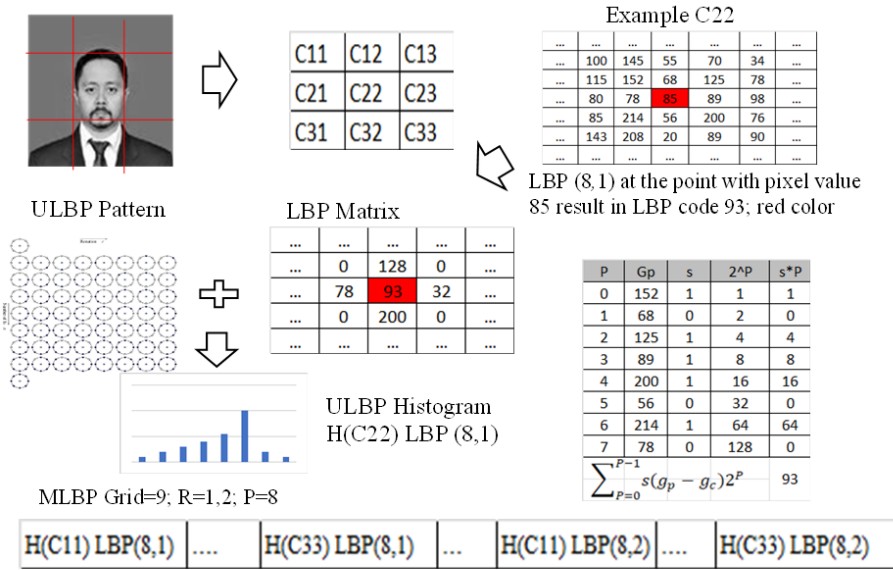

**Figure 4.** Illustration of the MLBP feature-extraction process.

The feature extraction using *LBP* begins by transforming all pixel information into *LBP* code. After generating the *LBP* code from all pixels, the next step is to generate a histogram-based feature vector by determining the appearance of the *LBP* code. The length of the feature vector from *LBP* depends on the number of neighbors processed, where the length of the feature vector is $2^P$. The formula for *LBP* is:

$$LBP_{P,R}(x_c, y_c) = \sum_{P=0}^{P-1} s(g_p - g_c) 2^P \quad s(x) = \begin{cases} 1, x \geq 0 \\ 0, x < 0 \end{cases} \tag{1}$$

where $P$ is the number of neighboring pixels, $R$ is the radius of the neighbors, $g_p$ is the gray level value of the neighboring pixels, $g_c$ is the gray level value of the processed pixel, and $s(x)$ is the threshold function. The *LBP* descriptor can be categorized as uniform and non-uniform patterns that can capture the microstructure of the image texture, such as the age spot region, edge, and corner. The binary pattern of a dot/pixel is expressed as a uniform pattern if the binary pattern contains at most two-bit-wise transitions on a circular binary string. Other patterns with larger-than-two-bit-wise changes are considered non-uniform. The resulting feature vector will be shorter by grouping the texture pattern into uniform and non-uniform. The pattern is divided into uniform and non-uniform because most of the local binary patterns in natural images are uniform, and uniform patterns are more

resistant from a statistical perspective (more resistant to noise). On the other hand, only considering a uniform pattern significantly decreases the number of possible *LBP* labels and is reliable.

### 3.2.2. Histogram of Gradient (HOG)

Histogram of gradient (HOG) is a handcrafted feature-extraction method based on the calculation of the appearance of a gradient in a specific orientation in a set of pixels of an image. The application of HOG in the recognition process has become well known and has been successfully applied in different recognition tasks such as facial expression, human detection, and object detection [6]. Figure 5 illustrates the HOG feature-extraction process.

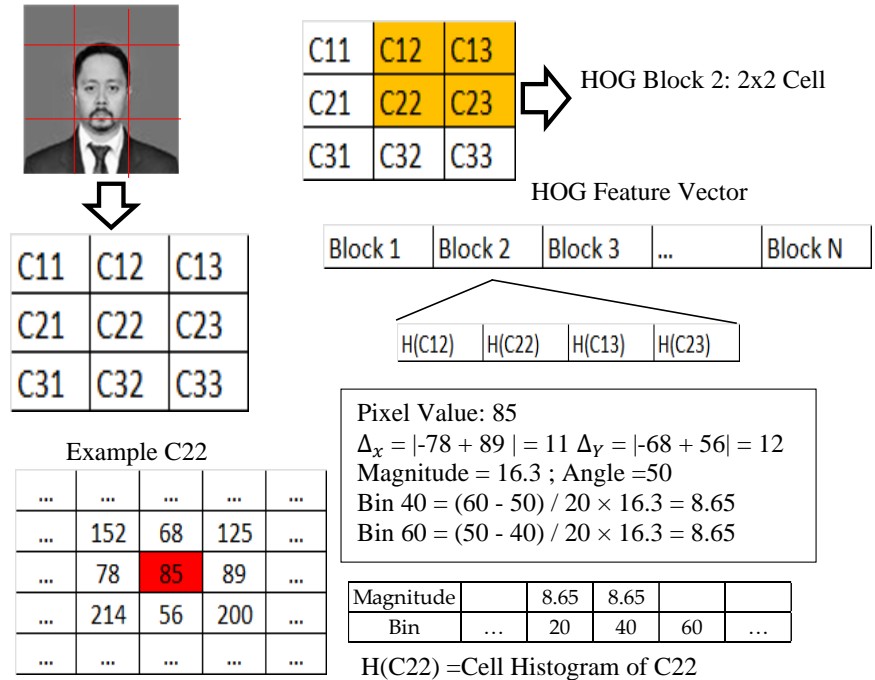

**Figure 5.** Illustration of the HOG feature-extraction process.

The HOG descriptor focuses on the distribution of local gradients or edges represented in the histogram. It ignores the location of local gradients or edges, which provides better generalization capabilities regarding the variance in the image. The HOG feature representation considers the gradients' magnitude value, where the distance between the orientation angle and the bin histogram angle becomes the basis for the magnitude value distribution. The equation for the gradient magnitude of a pixel at coordinate (*x,y*) is defined as follows:

$$\Delta_x = \left| G_{(x-1,y)} - G_{(x+1,y)} \right| \tag{2}$$

$$\Delta_y = \left| G_{(x,y-1)} - G_{(x,y+1)} \right| \tag{3}$$

$$M_{(x,y)} = \sqrt{\Delta_x^2 + \Delta_y^2} \tag{4}$$

$$\propto_{(x,y)} = \tan^{-1}\left(\frac{\Delta_y}{\Delta_x}\right) \tag{5}$$

where *G* is the grayscale value of a pixel in specific locations, $\Delta_x$ and $\Delta_y$ are the gradients in the horizontal and vertical orientation, $M_{(x,y)}$ is the gradient value at coordinate (*x,y*), and $\propto_{(x,y)}$ represents the gradient orientation.

In the initial stage of HOG, the first-order gradient on the normalized face image is calculated. Then, the image is divided into several cells, and the histogram formation process is carried out based on the orientation of the angle. The range of gradient angles used as a reference is determined beforehand. Next, gradient values at pixel positions are distributed based on the distance between the pixel orientation angle and the histogram orientation angle. Next, we overlap the cells to produce a block-based HOG descriptor. Then, the HOG descriptors from all blocks are combined to represent a feature vector.

### 3.2.3. Color Histogram (Color HIST)

Naturally, humans identify ethnicity using skin color as one of the phenotypic features. Therefore, feature representations based on color, such as the color histogram, are appropriate in the development of ethnicity classification tasks [22]. The determination of the occurrence of pixel intensity, normalization, and quantization makes up several processes to produce the color histogram to represent the color feature of an image [12]. The first-order histogram probability $P(g)$ is defined using Equation (6):

$$P(g) = \frac{N(g)}{M} \tag{6}$$

where $N(g)$ is the number of pixels at intensity value $g$, and $M$ is the number of pixels in the image. The value of $P(g)$ is less than or equal to 1, and the sum value equals 1. The number of bins from the histogram controls the size of the feature representation. Then, the histogram passes through normalization to overcome the difference in image dimensions, where the values in the histogram represent the probability of the color intensity. The loss of spatial information is a disadvantage of histogram-based representation. Color spatial features that quantize the pixel values from the sub-image area quantized into one representation value for the RGB channel contain artifacts of spatial information. Therefore, adding a spatial color feature can compensate for the loss of spatial information. Figure 6 illustrates the Color HIST feature-extraction process.

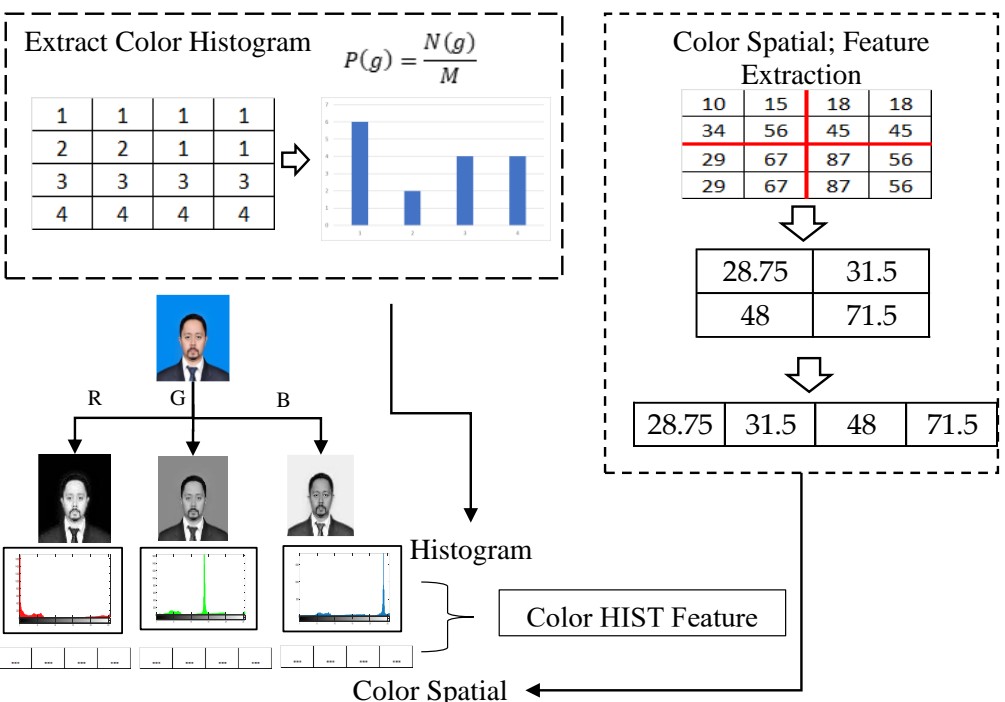

**Figure 6.** Illustration of the Color HIST feature-extraction process.

### 3.2.4. Speeded-up-Robust-Features-Based (SURF-Based)

Speeded-up robust feature (SURF) is a point-of-interest-based feature-extraction method that can produce robust features to scale and rotation variances. The SURF process consists of two main stages: detecting points of interest and describing the local environment [23]. The initial stage of SURF is to find interest points based on the Hessian matrix, where the application of box filters and integral images makes the computation process faster. Space scale is analyzed by increasing the filter size rather than decreasing the image size iteratively. After finding the interest points, the next step is to generate the SURF descriptor. It begins with the determination of the orientation using the Haar-wavelet, where the orientation with the most significant sum value indicates the primary orientation of the feature descriptor. For each point of interest, the descriptor is calculated by constructing a rectangular region with window sizes of 20s centered around the key point and oriented along the orientation. The region is split up regularly into smaller $4 \times 4$ square sub-regions. The summation and absolute values of the vertical and horizontal Haar-wavelet responses, $d_x$ and $d_y$, applied at $5 \times 5$ regularly spaced sample points, are computed to produce a SURF descriptor for each sub-region. Each sub-region has a four-dimensional descriptor vector, $v = (\Sigma d_x, \Sigma d_y, \Sigma |d_x|, \Sigma |d_y|)$, and the total number of descriptors for all $4 \times 4$ sub-regions is 64.

### 3.3. Independent Parameter Tuning

Independent parameter tuning became the initial stage in the proposed framework, with the aim of finding the grid-size parameter for each feature-extraction method, which achieves optimum conditions by increasing the feature size and level of accuracy. The feature-extraction process for MLBP and SURF-based features was performed on a grayscale image, while HOG and Color HIST feature extraction was performed on the RGB + gray channel. The MLBP, HOG, Color HIST, and SURF-based features are applied based on predetermined feature-extraction configuration to ensure the extracted feature vector has a similar size. The predetermined feature-extraction configuration is the parameter setting for feature extraction, for example, radius and number of neighbors in LBP, number of histogram bins and orientation in HOG, number of histogram bins and spatial color size in Color HIST, and scale values in SURF-based.

The feature-extraction method used in this study uses several parameter configuration variations to control the detailed level of the extraction process. Table 3 shows the parameter configuration variation used in the feature-extraction methods tested in this study. Based on Table 3, each feature-extraction method has four variations applied in the processed area. We used four variations; at first, the extraction results were two-dimensional due to parameter variations. Then, the features were rearranged into one-dimensional features. For example, the MLBP was applied in grayscale with $N = 8$ and four radius variations: 1, 3, 5, and 7. At first, the MLBP features were in two dimensions with dimensions of $59 \times 4$, resulting from extraction from eight neighbors at radius of 1, 3, 5, and 7. Then, the MLBP features were rearranged into one-dimensional features with dimensions of $1 \times 256$. A similar process was also applied for the HOG, Color HIST, and SURF-based features based on their parameter configurations.

The grid size divides the image into several sub-images that control the feature size and the details regarding the level of extraction in the image. For each sub-image, the MLBP, HOG, Color_Hist, and SURF-based feature-extraction methods are applied independently and concatenated to form features for the image level. The grid size also maintains the spatial location to rearrange feature vectors for a compact feature strategy. Finally, the grid size is tuned and evaluated using elbow analysis to find the optimum feature vector for each handcrafted method and the best optimization by increasing the feature size and accuracy level. The optimum feature is the extracted feature at a specific grid size, indicating optimization by increasing the feature size and accuracy level. The high-dimensional optimum feature results from independent parameter tuning contain redundant information due to the use of several parameter variations. Therefore, we further process the

optimum feature for each handcrafted method in the compact feature strategy stage. The independent parameter-tuning stage is described in Algorithm 1.

**Table 3.** Handcrafted feature-extraction configuration parameters.

| No. | Handcrafted Feature-Extraction Method | Configuration Parameters |
|---|---|---|
| 1 | Multi-Local Binary Pattern (MLBP) | Channel = {grayscale}; $N$ = 8; $R$ = {1,3,5,7}; Uniform Pattern; Feature size for one grid = 59 × 4 = 236 |
| 2 | Histogram of Gradient (HOG) | Channel = {[R, G, B, grayscale]}; bin = {9,18}; Block-size = {1 × 1}; Non-Overlap; Orientation = {[0, 180], [−180, 180]}; Feature size for one grid = 27 × 2 × 4 =216 |
| 3 | Color Histogram (Color HIST) | Channel = {[R, G, B, grayscale]}; bin = {32}; Color spatial info = {[6 × 6]}; Feature size for one grid = 32 × 4 + 108 =236 |
| 4 | Speeded-Up-Robust-Features-based (SURF-based) | Channel = {grayscale}; Point-based = {center of grid}; Scale = {1.6, 3.2, 4.8, 6.4}; Feature size for one grid = 64 × 4 =256 |

---

**Algorithm 1.** Compact-fusion feature framework: independent parameter tuning.

**PROCEDURE** Independent_Parameter_Tunning
        **(Input**:
        GS[1..ns]: array of grid-size
        ITrain [1..nt]: array of image {Training-set}
        IValidation [1..nv]: array of image {Validation-set}
        LTraining [1..nt]: array of label
        LValidation [1..nv]: array of label
        *{each ConfigFE contain n number of parameter for Feature Extraction}*
        ConfigFE [1..4]: array of struct configuration of Feature Extraction
        **Output:**
        Opt_Gs[1..4]: array of grid_size
        Train_Opt_MLBP, Val_Opt_MLBP: 2-D array of image feature
        Train_Opt_HOG, Val_HOG_MLBP: 2-D array of image feature
        Train_Opt_Color_HIST, Val_Opt_Color_HIST: 2-D array of image feature
        Train_Opt_SURF_based, Val_Opt_SURF_based: 2-D array of image feature
        **)**
**DECLARATION**
VR_MLBP, VR_HOG, VR_Color_HIST, VR_SURF_based: array of region feature
VI_MLBP, VI_HOG, VI_Color_HIST, VI_SURF_based: array of image feature
Train_VI_MLBP, Val_VI_MLBP: 3-D array of image feature
Train_VI_HOG, Val_VI_HOG: 3-D array of image feature
Train_VI_Color_HIST, Val_VI_Color_HIST: 3-D array of image feature
Train_VI_SURF_based, Val_VI_SURF_based: 3-D array of image feature
*{3-D array of image feature. 1-dimension indicates grid-size index used for feature extraction, 2-dimension indicates row/number of data, 3-dimension indicates kol/feature vectors}*
Acc_MLBP, Acc_HOG, Acc_Color_HIST, Acc_SURF_based: array of accuracy
Opt_Gs[1..4]: array of selected grid-size for optimum feature
ImageRegion: array of image {store splitting result}
Model_1, Model_2, Model_3,Model_4: SVM_Model
*{function/procedure used}*
Split_Image() *{function to split an image into sub-images based on grid size}*
MLBP_FE(), HOG_FE(), Color_HIST_FE(), SURF_based_FE() *{function to performed feature extraction}*
Concatenate() *{function to concatenate array}*
SVM_Train() *{function to train SVM Classifier}*
Evaluate() *{function to evaluate SVM Model}*
Elbow() *{function to generate data for manual elbow analysis}*

| Algorithm 1. *Cont.* |
|---|

**ALGORITHM**
**For** id_Grid ← 1 **to** ns
{processing Training-set}
      **For** j ← 1 **to** nt
         ImageRegion ← Split_Image(ITrain(j),GS(id_Grid))
         **For** x ← 1 **to** number_element_of(ImageRegion)
         *{ Using a predetermined configuration in ConfigFE, apply four independent handcrafted feature extraction methods, namely: MLBP, HOG, Color HIST, and SURF-based, to produce feature vectors on sub-Image named Vector Region (VR)}*
         VR_MLBP(x) ← MLBP_FE(ImageRegion(x),ConfigFE(1));
         VR_HOG(x) ← HOG _FE(ImageRegion(x),ConfigFE(2));
         VR_Color_HIST(x) ← Color_HIST_FE(ImageRegion(x),ConfigFE(3));
         VR_SURF_based(x) ← SURF_based_FE(ImageRegion(x),ConfigFE(4));
         **End**
         *{produce feature in image level by concatenating feature from sub-image}*
         VI_MLBP(j) ← concatenate(VR_MLBP)
         VI_HOG(j) ← concatenate(VR_HOG)
         VI_Color_HIST(j) ← concatenate(VR_Color_HIST)
         VI_SURF_based(j) ← concatenate(VR_SURF_based)
      **End**
*{store feature for training-set in 3-D array, id_Grid indicate the grid-size value used for feature extraction}*
Train_VI_MLBP(id_Grid,:,:) ← VI_MLBP
Train_VI_HOG(id_Grid,:,:) ← VI_HOG
Train_VI_Color_HIST(id_Grid,:,:) ← VI_Color_HIST
Train_VI_SURF_based(id_Grid,:,:) ← VI_SURF_based
*{processing Validation-set }*
      **For** j ← 1 **to** nv
         ImageRegion ← Split_Image(IValidation(j),GS(id_Grid))
         **For** x ← 1 **to** number_element_of(ImageRegion)
         *{ Using a predetermined configuration in ConfigFE, apply four independent handcrafted feature extracton methods, namely: MLBP, HOG, Color HIST, and SURF-based, to produce feature vectors on sub-Image named Vector Region (VR)}*
         VR_MLBP(x) ← MLBP_FE(ImageRegion(x),ConfigFE(1));
         VR_HOG(x) ← HOG _FE(ImageRegion(x),ConfigFE(2));
         VR_Color_HIST(x) ← Color_HIST_FE(ImageRegion(x),ConfigFE(3));
         VR_SURF_based(x) ← SURF_based_FE(ImageRegion(x),ConfigFE(4))
         **End**
         *{produce feature in image level by concatenating feature from sub-image}*
         VI_MLBP(j) ← Concatenate(VR_MLBP)
         VI_HOG(j) ← Concatenate(VR_HOG)
         VI_Color_HIST(j) ← Concatenate(VR_Color_HIST)
         VI_SURF_based(j) ← Concatenate(VR_SURF_based)
      **End**
*{store feature for training-set in 3-D array, id_Grid indicate the grid-size value used for feature extraction}*
Val_VI_MLBP(id_Grid,:,:) ← VI_MLBP
Val_VI_HOG(id_Grid,:,:) ← VI_HOG
Val_VI_Color_HIST(id_Grid,:,:) ← VI_Color_HIST
Val_VI_SURF_based(id_Grid,:,:) ← VI_SURF_based
*{feature vector from Training-set and Validation-set for GS(i) are produced }*
*{Train SVM Classifiers for each handcrafted feature }*
Model_1 ← SVM_Train(Train_VI_MLBP(id_Grid,:,:), LTraining)
Model_2 ← SVM_Train(Train_VI_HOG(id_Grid,:,:), LTraining)
Model_3 ← SVM_Train(Train_VI_Color_HIST(id_Grid,:,:), LTraining)
Model_4 ← SVM_Train(Train_VI_SURF_based(id_Grid,:,:), LTraining)

| **Algorithm 1.** *Cont.* |
| --- |
| *{Calculate accuracy from Validation-set }*<br>Acc_MLBP(id_Grid) ← Evaluate(Model_1, Val_VI_MLBP(i), LValidation)<br>Acc_HOG(id_Grid) ← Evaluate(Model_2, Val_VI_HOG(i), LValidation)<br>Acc_Color_HIST(id_Grid) ← Evaluate(Model_3, Val_VI_Color_HIST(i), LValidation)<br>Acc_SURF_based(id_Grid) ← Evaluate(Model_4, Val_VI_SURF_based(i), LValidation)<br>**End**<br>*{Using accuracy from Validation-set, find the optimum feature for each handcrafted feature extraction method using elbow analysis. The optimum feature is a vector at GS(id_Grid) that provides an optimum condition which indicates an optimization between increasing feature size and accuracy}*<br>[Train_Opt_MLBP, Val_Opt_MLBP, Opt_Gs(1)] ← Elbow(Acc_MLBP, GS)<br>[Train_Opt_HOG, Val_HOG_MLBP, Opt_Gs(2)] ← Elbow(Acc_ HOG, GS)<br>[Train_Opt_Color_HIST, Val_Opt_Color_HIST, Opt_Gs(3)] ← Elbow(Acc_Color_HIST, GS)<br>[Train_Opt_SURF_based, Val_Opt_SURF_based, Opt_Gs(4)] ← Elbow(Acc_SURF_based, GS) |

*3.4. Compact Feature Strategy*

The compact feature strategy stage produces features that minimize redundant information from an optimum feature, as seen in Figure 7. The proposed compact feature strategy designs a single data method, which only needs a feature vector from a single processed image. This is achieved by selecting or generating a new presentation using only one row of the feature vector information from an optimum feature for each type of feature-extraction method.

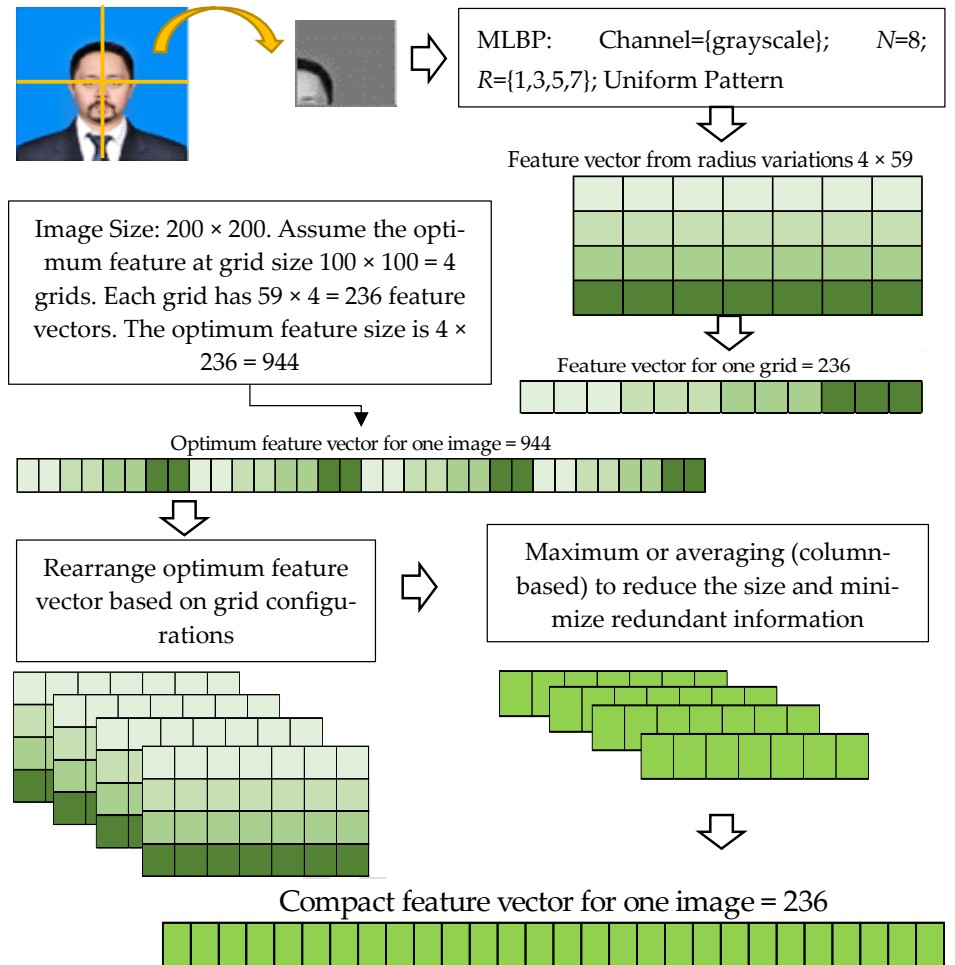

**Figure 7.** Illustration of feature transformation from optimum feature to compact feature for MLBP feature.

The proposed compact feature method is as follows: First, the optimum feature is rearranged based on the spatial location; this can be achieved because the grid-based approach defines the feature sizes for each handcrafted feature and has exact feature sizes for each handcrafted feature. Then, the compact feature strategy is applied to the rearranged optimum feature. The selection strategy uses the maximum value method, adapted from [16–18], using the formula defined in Equation (7). Meanwhile, a new presentation is generated by calculating the average value of feature vectors, adapted from [1,20], based on the equation defined in Equation (8).

$$\widehat{H_r} = argmax\sum\nolimits_{i=1}^{n} Hi_r \tag{7}$$

$$\widehat{H_r} = \frac{\sum_{i=1}^{n} Hi_r}{n} \tag{8}$$

$$\hat{H} = \bigcup\nolimits_{r=1}^{k} \widehat{H_r} \tag{9}$$

where $\hat{H}_r$ is the compact feature at grid number $r$, $n$ is the number of feature vector configurations applied, and $Hi_r$ is the feature vector for $i$ configuration at grid number $r$. The compact feature of the image $\hat{H}$ is produced by concatenating the compact feature from the grids $\hat{H}_r$ into a one-dimensional vector. The maximum value method selects the feature vector with the maximum value, considering that the maximum value is intended to provide peak feature information that provides better variation. The strategy to generate new presentations aims to accumulate new presentations by averaging the feature information from rearranged two-dimensional feature vectors to produce a smaller compact feature representation. In ideal conditions, the compact feature is generated from the optimum feature by selecting the feature-reduction methods that achieve higher accuracy and minimum accuracy decrement. The compact feature strategy stage is described in Algorithm 2.

---

**Algorithm 2.** Compact-fusion feature framework: compact feature strategy.

---

**PROCEDURE** Compact_Feature
    **(Input**:
    Opt_Gs [1..4]: GS that is considered optimum for each Feature Extraction
    { 2-D array of image features that are considered optimum features }
    Train_Opt_MLBP, Val_Opt_MLBP: 2-D array of image feature
    Train_Opt_HOG, Val_Opt_HOG: 2-D array of image feature
    Train_Opt_Color_HIST, Val_Opt_Color_HIST: 2-D array of image feature
    Train_Opt_SURF_based, Val_Opt_SURF_based: 2-D array of image feature
    ConfigFE [1..4]: array of struct configuration of Feature Extraction
    GS[1..ns]: array of grid-size
    LTraining [1..nt]: array of label
    LTValidation [1..nv]: array of label
    **Output:**
    Cmp_S: selected compact strategies
    Train_Cmp_MLBP, Val_Opt_MLBP: 2-D array of image feature
    Train_Cmp_HOG, Val_HOG_MLBP: 2-D array of image feature
    Train_Cmp_Color_HIST, Val_Opt_Color_HIST: 2-D array of image feature
    Train_Cmp_SURF_based, Val_Cmp_SURF_based: 2-D array of image feature
    **)**

---

---

**Algorithm 2.** *Cont.*

---

**DECLARATION**

VR_MLBP, VR_HOG, VR_Color_HIST, VR_SURF_based: 2-D array of region feature
VI_MLBP, VI_HOG, VI_Color_HIST, VI_SURF_based: array of struct image feature
Train_VI_MLBP, Val_VI_MLBP: 2-D array of image feature
Train_VI_HOG, Val_VI_HOG: 2-D array of image feature
Train_VI_Color_HIST, Val_VI_Color_HIST: 2-D array of image feature
Train_VI_SURF_based, Val_VI_ URF_based: 2-D array of image feature
Acc_MLBP, Acc_HOG, Acc_Color_HIST, Acc_SURF_based: array of struct for accuracy
*{function/procedure used}*
rearrange() *{function to rearrange optimum feature to 2-D array}*
selectMax() *{function to calculate maximum in column-based 2-D array}*
selectAverage() *{function to calculate mean in column-based 2-D array}*
SVM_Train() *{function to train SVM Classifier}*
Evaluate() *{function to evaluate SVM Model}*
Analyse() *{function for accuracy analysis}*
**ALGORITHM**
{Extract Compact Feature from Optimum Feature on Training-set}
**For** j ← 1 **to** nt
*{rearrange the Optimum feature into a 2-D array based on the optimum grid size and configuration of Feature Extraction. The illustration can be seen in*Figure 7*, continued by applying two strategies: maximum selecting method and averaging value method for each data}*
VR_MLBP(j) ← rearrange(Train_Opt_MLBP(j),GS(Opt_Gs(1)), ConfigFE(1))
VI_MLBP(j).Max ← selectMax(VR_MLBP(j))
VI_MLBP(j).Mean ← selectAverage(VR_MLBP(j))
VR_HOG(j) ← rearrange (Train_Opt_HOG(j),GS(Opt_Gs(2)), ConfigFE(2))
VI_HOG(j).Max ← selectMax(VR_HOG (j))
VI_HOG(j).Mean ← selectAverage(VR_HOG(j))
VR_Color_HIST(j) ← rearrange (Train_Opt_Color_HIST(j),GS(Opt_Gs(3)),ConfigFE(3))
VI_Color_HIST (j).Max ← selectMax(VR_Color_HIST (j))
VI_ Color_HIST (j).Mean ← selectAverage(VR_Color_HIST (j))
VR_SURF_based(j) ← rearrange(Train_Opt_SURF_based(j),GS(Opt_Gs(4)), ConfigFE(4))
VI_SURF_based(j).Max ← selectMax(VR_SURF_based(j))
VI_SURF_based (j).Mean ← selectAverage(VR_SURF_based(j))
**End**
*{store feature for training-set in variable}*
Train_VI_MLBP ← VI_MLBP
Train_VI_HOG ← VI_HOG
Train_VI_Color_HIST ← VI_Color_HIST
Train_VI_SURF_based ← VI_SURF_based
*{Extract Compact Feature from Optimum Feature on Training-set}*
**For** j ← 1 **to** nv
*{rearrange the Optimum feature into a 2-D array based on the optimum grid size and configuration of Feature Extraction. The illustration can be seen in*Figure 7*, continued by applying two strategies: maximum selecting method and averaging value method for each data}*
VR_MLBP(j) ← rearrange(Val_Opt_MLBP(j),GS(Opt_Gs(1)), ConfigFE(1))
VI_MLBP(j).Max ← selectMax(VR_MLBP(j))
VI_MLBP(j).Mean ← selectAverage(VR_MLBP(j))
VR_HOG(j) ← rearrange (Val_Opt_HOG(j),GS(Opt_Gs(2)), ConfigFE(2))
VI_HOG (j).Max ← selectMax(VR_HOG (j))
VI_HOG (j).Mean ← selectAverage(VR_HOG (j))
VR_Color_HIST(j) ← rearrange (Val_Opt_Color_HIST(j),GS(Opt_Gs(3)),ConfigFE(3))
VI_Color_HIST (j).Max ← selectMax(VR_Color_HIST(j))
VI_Color_HIST (j).Mean ← selectAverage(VR_Color_HIST(j))
VR_SURF_based(j) ← rearrange (Val_Opt_SURF_based(j),GS(Opt_Gs(4)), ConfigFE(4))

---

**Algorithm 2.** *Cont.*

---

VI_SURF_based(j).Max ← selectMax(VR_SURF_based (j))
VI_SURF_based(j).Mean ← selectAverage(VR_SURF_based (j))
**End**
*{store compact feature for validation-set in variable}*
Val_VI_MLBP ← VI_MLBP
Val_VI_HOG ← VI_HOG
Val_VI_Color_HIST ← VI_Color_HIST
Val_VI_SURF_based ← VI_SURF_based
*{Train SVM Classifiers with the compact feature from maximum method}*
Model_1 ← SVM_Train(Train_VI_MLBP.Max, LTraining)
Model_2 ← SVM_Train(Train_VI_HOG.Max, LTraining)
Model_3 ← SVM_Train(Train_VI_Color_HIST(i).Max, LTraining)
Model_4 ← SVM_Train(Train_VI_SURF_based(i).Max, LTraining)
*{Calculate accuracy from Validation-set}*
Acc_MLBP.Max ← evaluate(Model_1, Val_VI_MLBP.Max)
Acc_HOG.Max ← evaluate(Model_2, Val_VI_HOG.Max)
Acc_Color_HIST.Max ← evaluate(Model_3, Val_VI_Color_HIST.Max)
Acc_SURF_based.Max ← evaluate(Model_4, Val_VI_SURF_based.Max)
*{Train SVM Classifiers with the compact feature from averaging method}*
Model_1 ← SVM_Train(Train_VI_MLBP.Mean, LTraining)
Model_2 ← SVM_Train(Train_VI_HOG.Mean, LTraining)
Model_3 ← SVM_Train(Train_VI_Color_HIST(i).Mean, LTraining)
Model_4 ← SVM_Train(Train_VI_SURF_based(i).Mean, LTraining)
*{Calculate accuracy from Validation-set}*
Acc_MLBP.Mean ← Evaluate(Model_1, Val_VI_MLBP.Mean)
Acc_HOG.Mean ← Evaluate(Model_2, Vali_VI_HOG.Mean)
Acc_Color_HIST.Mean ← Evaluate(Model_3, Val_VI_Color_HIST.Mean)
Acc_SURF_based.Mean ← Evaluate(Model_4, Val_VI_SURF_based.Mean)
*{Using accuracy from Validation-set, evaluate the accuracy of candidate compact features compared with optimum features and determine the compact method used }*
[Train_Cmp_MLBP,Val_Cmp_MLBP,Train_Cmp_HOG,Val_Cmp_HOG_MLBP,
Train_Cmp_Color_HIST,Val_Cmp_Color_HIST,Train_Cmp_SURF_based,
Val_Cmp_SURF_based,Cmp_S]←Analyse(Acc_MLBP,
Acc_HOG,Acc_Color_HIST,Acc_Color_HIST)

---

*3.5. Fusion at The Feature Level*

The compact feature strategy results in features with smaller sizes than the optimum features, but the method suffered from classification performance degradation. It should be noted that the independent parameter tuning and compact feature strategy stages that are carried out on the MLBP, HOG, Color HIST, and SURF-based features are evaluated independently (single feature). Finally, the compact-fusion feature is produced by fusing compact features of the MLBP, HOG, Color HIST, and SURF-based features to produce a multi-feature that is able to counter performance degradation problems [1,11,12]. The feature fusion stage is described in Algorithm 3.

---

**Algorithm 3.** Compact-fusion feature framework: feature fusion at the feature level.

---

**PROCEDURE** Feature_Fusion
   **(Input:**
   Cmp_S: selected compact strategies
   *{2-D array of image features that are considered compact features }*
   Train_Cmp_MLBP, Val_Cmp_MLBP: 2-D array of image feature
   Train_Cmp_HOG, Val_Cmp_HOG: 2-D array of image feature
   Train_Cmp_Color_HIST, Val_Cmp_Color_HIST: 2-D array of image feature

---

---

**Algorithm 3.** *Cont.*

---

    Train_Cmp_SURF_based, Val_Cmp_SURF_based: 2-D array of image feature
    LTraining [1..nt]: array of label
    LTValidation [1..nv]: array of label
    *{from independent parameter tunning }*
    Opt_Gs [1..4]: GS that consider optimum for each Feature Extraction
    ConfigFE [1..4]: array of struct configuration of Feature Extraction
    Opt_Gs [1..4]: GS that is considered optimum for each Feature Extraction
    **Output:**
    Config_CF: struct of compact feature configuration
    Model_CF: SVM Model {best SVM Model}
    **)**
**DECLARATION**
Train_CF, Val_CF: 2-D array of image feature
Acc_CF: variable to save the accuracy
Config_CF: struct of compact feature configuration
*{function/procedure used}*
Concatenate() *{function to concatenate array}*
SVM_Train() *{function to train SVM Classifier}*
Evaluate() *{function to evaluate SVM Model}*
Analyse() *{function for accuracy analysis}*
**ALGORITHM**
*{Concatenate Compact Feature of Training-set}*
**For** j ← 1 **to** nt
*{concatenate four optimum compact features MLBP, HOG, Color_HIST, SURF_based}*
    Train_CF (j) ← Concatenate(Train_Cmp_MLBP(j), Train_Cmp_HOG(j),
    Train_Cmp_Color_HIST(j), Train_Cmp_SURF_based(j))
**End**
*{Concatenate Compact Feature of Validation-set}*
**For** j ← 1 **to** nv
*{ concatenate four optimum compact features MLBP, HOG, Color_HIST, SURF_based}*
    Val_CF (j) ← Concatenate(Val_Cmp_MLBP(j), Val_Cmp_HOG(j),
    Val_Cmp_Color_HIST(j), Val_Cmp_SURF_based(j))
**End**
*{Train SVM Classifiers with compact-fusion feature}*
Model_1 ← SVM_Train(Train_CF, LTraining)
*{Calculate accuracy from Validation-set to determine the best model in case there were several model build}*
Acc_CF ← evaluate(Model_1, Val_CF)
*{Model from SVM Classifier and best configuration of Compact Fusion to used for the testing dataset}*
[Model_CF, Config_CF] ← Analyse(Acc_CF)

---

## 4. Experiment and Results

This section describes the seven stages conducted to develop and test the proposed framework: (1) the dataset and experimental preparation and setup, (2) the initial experiment results and discussion, (3) the ablation study, (4) the comparison of the proposed framework with proven feature dimension methods, (5) cross-dataset experiments, (6) the classifier comparison, and (7) the comparison of the proposed method with SOTA.

### 4.1. Dataset and Experimental Preparation

The datasets selected for this study were the UTKFace [24] and Fair Face datasets [25]. Our choice of datasets considered the number of images, public availability, number of ethnicity groups, variations in image quality, data distribution, and availability of other research results. The UTKFace dataset was used for the experiment in which we tuned the parameters of the proposed method. Meanwhile, the Fair Face dataset was used in the cross-dataset experiment to analyze the generalization of the parameters of the proposed method.

The UTKFace dataset can be accessed at https://susanqq.github.io/UTKFace/ (accessed on 30 May 2023), and it consists of over 20,000 face images with annotations of age, gender, and ethnicity. In addition, the dataset covers significant variations in pose, facial expression, illumination, occlusion, and resolution [24].

The Fair Face dataset is balanced based on race, consisting of 108,501 images from 7 races: White, Black, Indian, East Asian, Southeast Asian, Middle Eastern, and Latino [2,3]. Fair Face [25] can be accessed at https://github.com/joojs/fairface/ (accessed on 30 May 2023), and contains challenging appearance, illumination, and pose variation due to the images being taken from bad angles in unconstrained environments. The study by Belcar et al. [2] reported that the preprocessing stage to detect landmark points failed in a high number of image data on the Fair Face dataset. This indicates that there is a high number of non-frontal face images in the Fair Face dataset.

Figure 8 shows that the variations in appearance, in terms of age, gender, and five racial classes: White, Black, Asian, Indian, and Others (such as Hispanic, Latino, and Middle Eastern) are very high in both datasets due to data acquisition taking place in an unconstrained environment. The appearance variation in the unconstrained-environment dataset causes a high degree of skin color bias, which is a visual phenotypic trait used to identify ethnicity.

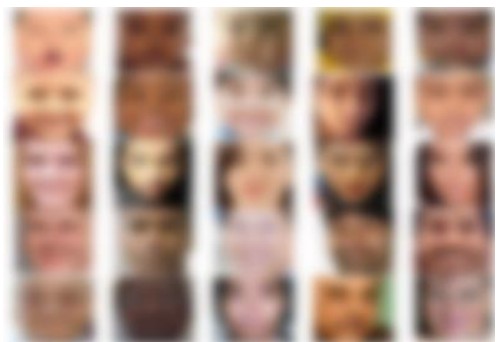 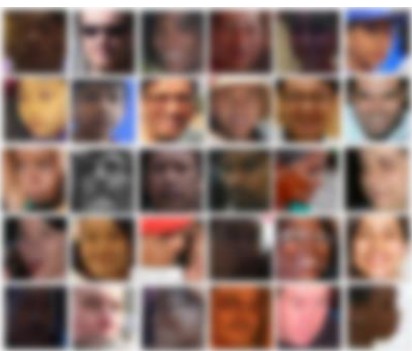

**Figure 8.** Examples of the images from the UTKFace dataset (**left**) and the Fair Face dataset (**right**); the images are blurred due to privacy concerns.

### 4.1.1. Dataset for Initial Experiment

For the initial experiment, this study used the aligned and cropped-face version from the UTKFace dataset; it consists of 23,750 face images of different people aged 1 to 116 years divided into five racial classes, which are: White, Black, Asian, Indian, and Others (such as Hispanic, Latino, and Middle Eastern). The selected datasets have ethnicity labels for each image, serving as ground truth values for the calculation of performance measurements. In addition, face detection is applied as a preprocessing process to ensure that each image from the dataset contains a front-facing human face. Then, after confirming the presence of a face in the image, the images are normalized into 200 × 200-pixel sizes and saved as a preprocessing dataset. Meanwhile, the image data that fail to be detected as a front-facing human face are excluded from the preprocessing dataset. Therefore, from the total original data of 23,750, after preprocessing, the number of data becomes 22,332. The original data distribution is accepted as balanced, with percentages of composition for White, Black, Asian, Indian, and Others being 43%, 19%, 14%, 17%, and 7%, respectively. The preprocessing results do not significantly change the distribution of ethnicity labels; the data distribution becomes 43%, 19%, 14%, 17%, and 7% for White, Black, Asian, Indian, and Others. Therefore, after preprocessing, the UTKFace dataset should still provide exceptional data quality for the evaluated ethnicity classification algorithms. Figure 9 compares all ethnicities represented in the original UTKFace dataset against the preprocessing UTKFace dataset used in this study.

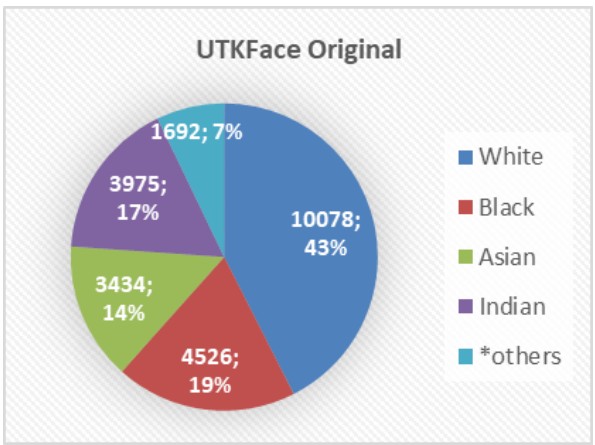
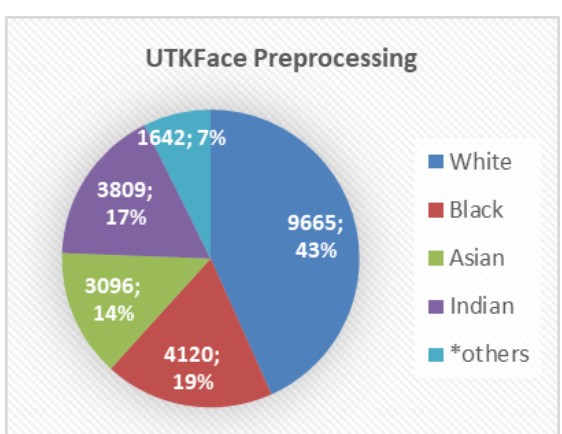

**Figure 9.** Comparison of original UTKFace vs. preprocessing UTKFace datasets. * contains Hispanic, Latino, and Middle Eastern ethnicity.

### 4.1.2. Dataset for Cross-Dataset Experiment

The cross-dataset experiment aims to evaluate the performance of the best configuration of the compact-fusion feature on other datasets. The dataset used in the cross-dataset experiment is a subset of the Fair Face dataset. The Fair Face dataset is a balanced dataset based on race, consisting of 108,501 images from seven races: White, Black, Indian, East Asian, Southeast Asian, Middle Eastern, and Latino [3,25]. Due to the difference in the number of ethnic labels between the UTKFace and Fair Face datasets, this test uses the four ethnic classes that are present in both datasets: White, Black, Asian, and Indian.

The UTKFace Preprocessing dataset, which is reduced to only four classes, is named UTKFace Preprocessing-A and consists of 20,690 data divided into 15,522 training data and 5168 testing data. The ethnicity distribution of the data in UTKFace Preprocessing-A is as follows: 47% White, 20% Black, 15% Asian, and 18% Indian. Meanwhile, the Fair Face dataset used in this test is a subset called Fair Face Subset-A, consisting of 18,963 data, including 14,218 training data and 4745 testing data. Face detection preprocessing is performed on Fair Face Subset-A to ensure the faces are front-facing. As a result, the ethnicity distribution of the data in Fair Face Subset-A is as follows: 32% White, 20% Black, 25% Asian, and 23% Indian. The comparison is shown in Figure 10.

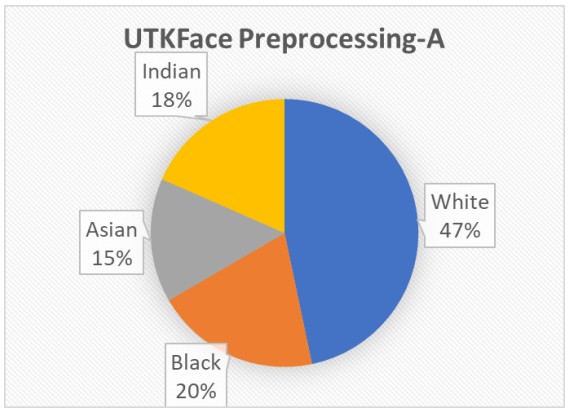
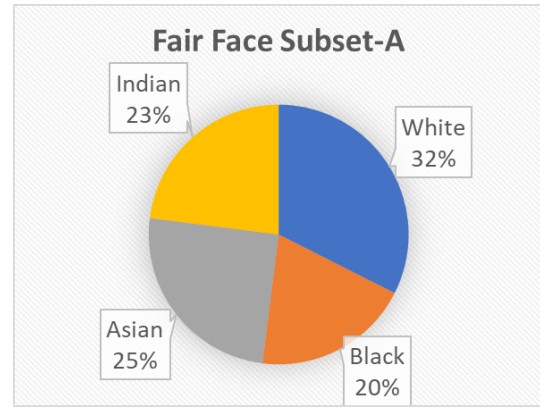

**Figure 10.** The comparison of ethnicities in UTKFace Preprocessing-A (**left**) and Fair Face Subset-A (**right**).

### 4.1.3. Experimental Preparation and Setup

The initial experiment applies the holdout validation protocol that divides the dataset into training, validation, and testing sets, with the proportion of 50%, 25%, and 25%, respectively. The dividing process uses stratified random sampling to ensure variation in the sample, and each subset has similar sample variations. The final data used in this study consist of 22,332 data split into 11,183 in the training set, 5572 in the validation set, and 5577 in the testing set. In addition, the experiment uses the training set for the feature-extraction configuration and learning classification model. Next, the validation set is used to adjust the configuration and classification model to achieve the best performance. Finally, the experiment in the testing set will evaluate the capability to generalize the feature extraction and classification model. The classifier in the experiment uses the SVM multi-class and implemented OVA schemes (One Versus All). The learning parameter used for the initial experiments is a polynomial kernel with order 3, C = 1, and the kernel scale is set as auto (which uses a heuristic procedure to select the scale value with a random number set to 5); this parameter is based on initial observations using several different kernels (linear, quadratic, polynomial, and RBF).

The measurement parameters in the experiments are adjusted according to the analysis needs, which include: classification accuracy, dimension reduction ratio, feature size, precision, recall, and F1-Score, using Equations (10)–(14).

$$Accuracy = \frac{True\_Positive + True\_Negative}{True\_Positive + True\_Negative + False\_Positive + False\_Negative} \tag{10}$$

$$Reduction\ ratio = 1 - \frac{reduction\ size}{original\ size} \tag{11}$$

$$Precision = \frac{True\_Positive}{True\_Positive + False\_Positive} \tag{12}$$

$$Recall = \frac{True\_Positive}{True\_Positive + False\_Negative} \tag{13}$$

$$F1\ score = 2\ *\ \frac{Precision * Recall}{Precision + Recall} \tag{14}$$

### 4.2. Initial Experiment Result and Discussion

This study aims to design compact and efficient features for ethnicity classification through a comprehensive initial experiment consisting of independent parameter tuning, a compact feature strategy, and a fusion strategy. The experiment begins by independently evaluating the accuracy performance of each handcrafted feature; it attempts to find the grid-size value that provides equality between accuracy and feature size. However, the optimum feature still contains redundant information caused by multiple variations in the feature-extraction process. Next, compact features are extracted from optimum features using maximum and average values. As a result, the compact feature will be smaller than the optimum features but has a lower discriminant capability. After that, the fusion strategy is applied to reach a compromise between the size and accuracy of the ethnicity classification task.

The independent parameter tuning, compact feature strategy, and fusion-at-the-feature-level experiment use a testing and validation set to provide the best configuration and learning model to be evaluated in the testing set. It should be noted that the independent parameter tuning and compact feature strategy stages are evaluated independently (single feature). Finally, the experiment is conducted in the testing dataset to show the comparative performance of the proposed framework compared with a single-feature approach and fusion without a compact process.

### 4.2.1. Independent Parameter-Tuning Experiment

The independent parameter-tuning experiment separately evaluates (treated as a single feature) each feature's performance to produce the optimum feature through accuracy analysis to determine the equilibrium position, which indicates an optimization between increasing feature size and accuracy. The data to find the equilibrium position are generated by changing the grid size, which impacts the detail and size of the extracted feature for all feature-extraction parameters used. The experiment uses five grid sizes: $200 \times 200$, $100 \times 100$, $50 \times 50$, $40 \times 40$, and $25 \times 25$. The results of independent parameter tuning are shown in Table 4.

**Table 4.** Results of parameter tuning for each handcrafted feature.

| Grid Size | MLBP | | HOG | | Color HIST | | SURF-Based | |
|---|---|---|---|---|---|---|---|---|
| | Size | Acc. | Size | Acc. | Size | Acc. | Size | Acc. |
| $200 \times 200$ | 236 | 57.09% | 216 | 40.47% | 236 | 65.11% | 256 | 71.28% |
| $100 \times 100$ | 944 | 67.43% | 864 | 54.88% | 944 | 73.22% | 1024 | 77.35% |
| $50 \times 50$ | 3776 | 76.24% | 3456 | 69.63% | 3776 | 76.85% | 4096 | 78.32% |
| $40 \times 40$ | 5900 | 78.28% | 5400 | 72.54% | 5900 | 76.92% | 6400 | 77.96% |
| $25 \times 25$ | 15,104 | 80.20% | 13,824 | 77.39% | 15,104 | 77.51% | 16,384 | 77.39% |

In theory, the larger the feature vector size, the greater the data variance, which increases the possibility of separating data with different characteristics. The best level of accuracy achieved from independent parameter tuning is 80.96%, 77.39%, 77.51%, and 78.32%, respectively, for the MLBP, HOG, Color HIST, and SURF-based features, with a feature size of 15,104, 13,824, 15,104, and 4096. The experimental results in Table 4 show a directly proportional relationship between feature size and accuracy, except for SURF-based features, which achieve a peak accuracy level at 4096 features. However, there is a saturated condition in which the increment in feature size is not comparable with the increment in accuracy, as seen in Figure 11. Elbow analysis from the experiment results indicates two optimum candidates for grid parameters, denoting a saturation condition for each feature. Therefore, two optimum candidates are selected for further investigation and analysis regarding the relationship between feature size and accuracy, named Optimum-1 and Optimum-2 features.

At this stage, the already-obtained optimum feature for each handcrafted feature, which is the Optimum-1 feature with a smaller feature size, consists of MLBP [50 × 50], HOG [40 × 40], Color HIST [50 × 50], and SURF [100 × 100]. Meanwhile, the Optimum-2 feature consists of MLBP [40 × 40], HOG [25 × 25], Color HIST [40 × 40], and SURF-based [50 × 50] features. Two optimum features will be processed further at the compact feature strategy stage.

### 4.2.2. Compact Feature Strategy Experiment

The grid parameter tuning produces two optimum features that mark the equilibrium between feature size and accuracy. Although the optimum feature achieves an acceptable minimum accuracy level of ~70%, it has a high dimension and creates redundant information that is caused by variations in the channel and extraction configuration. Therefore, the next step is to minimize redundant information from the optimum feature to produce a compact feature with a smaller size while maintaining accuracy through feature selection. The feature-selection method performs an element-based operation for spatial locations that contain redundant information. The maximum value and simple averaging methods are applied to minimize redundant information. The process is applied as single-row feature vector operations that provide trace-back characteristics for an explanation if needed.

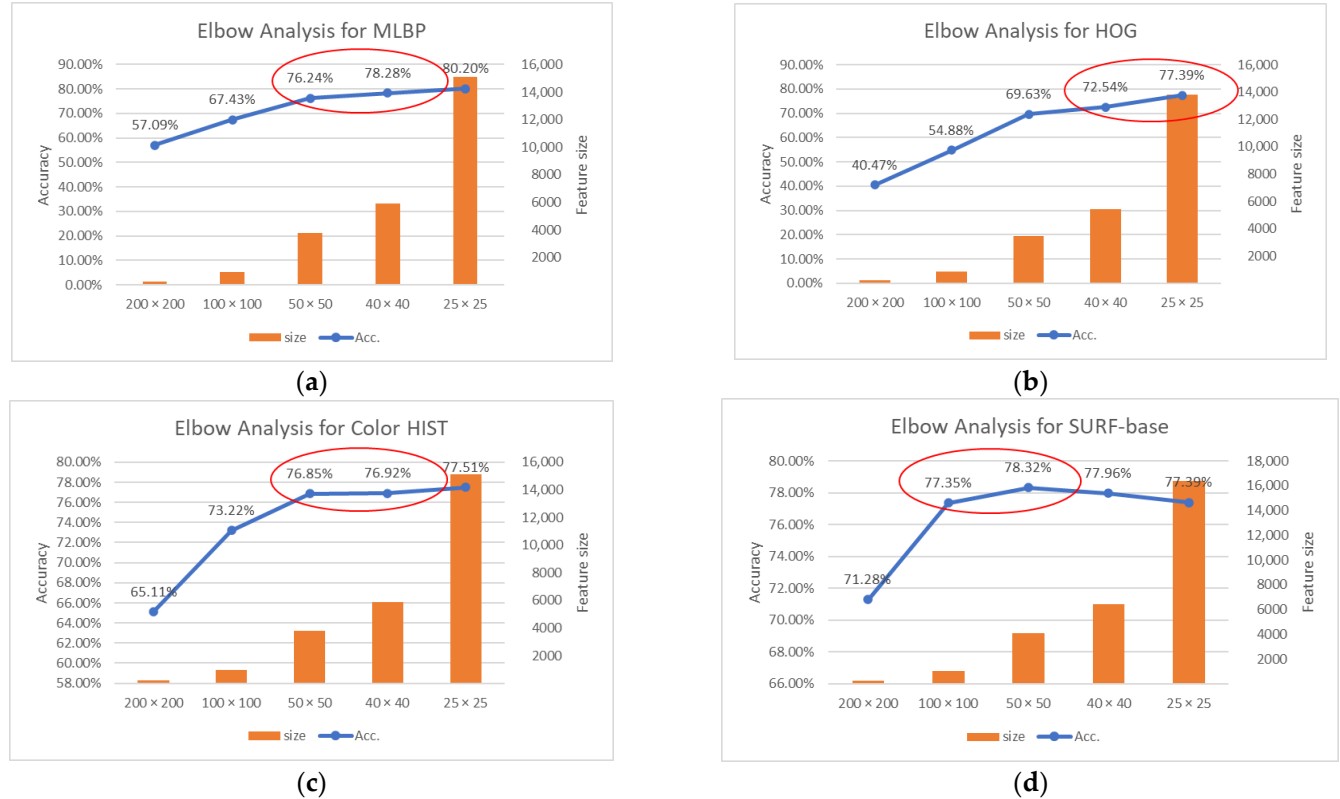

**Figure 11.** Elbow analysis for (**a**) MLBP, (**b**) HOG, (**c**) Color HIST, and (**d**) SURF-based features. Thr red circle indicated two optimum candidates.

From the predetermined parameter in Table 3 in Section 3.3, the handcrafted feature will have four variations for each grid location. This means that the maximum value and simple averaging methods are applied in four values for each attribute to produce one value for each attribute, resulting in an estimated 75% ratio reduction. Except for the Color HIST, the spatial color features that contain spatial color distribution are excluded in the compact feature process. The results of the compact feature strategy are shown in Table 5.

**Table 5.** Results of compact feature strategy experiment.

| Optimum ID | Feature Type | Optimum Feature | | Compact Feature | | | Delta Acc. (%) | | Reduction Ratio (%) |
|---|---|---|---|---|---|---|---|---|---|
| | | Size [a] | Acc. (%) [b] | Size [c] | Avg. Acc. (%) * [d] | Max Acc. (%) ** [e] | [b]–[d] | [b]–[e] | 1-[c]/[a] |
| 1 | MLBP [50 × 50] | 3776 | 76.24 | 944 | 75.22 | 73.38 | 1.02 | 2.86 | 75.00 |
| 1 | HOG [40 × 40] | 5400 | 72.54 | 1350 | 68.19 | 70.42 | 4.35 | 2.12 | 75.00 |
| 1 | Color HIST [50 × 50] | 3776 | 76.85 | 2240 | 78.21 | 77.82 | −1.36 | −0.97 | 40.68 |
| 1 | SURF-based [100 × 100] | 1024 | 77.35 | 256 | 70.70 | 68.72 | 6.65 | 8.63 | 75.00 |
| Average of Optimum-1 group | | - | 75.75 | - | 73.08 | 72.59 | 2.67 | 3.16 | 66.42 |
| 2 | MLBP [40 × 40] | 5900 | 78.28 | 1475 | 77.44 | 75.95 | 0.84 | 2.33 | 75.00 |
| 2 | HOG [25 × 25] | 13,824 | 77.39 | 3456 | 72.30 | 76.60 | 5.09 | 0.79 | 75.00 |
| 2 | Color HIST [40 × 40] | 5900 | 76.92 | 3500 | 79.33 | 78.18 | −2.41 | −1.26 | 40.68 |
| 2 | SURF-based [50 × 50] | 4096 | 78.32 | 1024 | 77.34 | 74.82 | 0.98 | 3.5 | 75.00 |
| Average of Optimum-2 group | | | 77.54 | | 76.60 | 76.39 | 1.13 | 1.34 | 66.42 |

\* Denotes accuracy using averaging method; ** denotes accuracy using maximum value method.

By analyzing the experimental results in Table 5, we can see that overall, the compact feature produced from the Optimum-1 group and Optimum-2 group are ~3 times smaller features than the original (reduction ratio at ±67%). Furthermore, by analyzing the data in Table 5, we can see that, on average, the compact feature from the Optimum-2 group, referred to as Compact-Optimum-2, results in better performance compared with the compact feature from the Optimum-1 group, now called Compact-Optimum-1. For Compact-Optimum-2 and Compact-Optimum-1, the average accuracy level is 76.60%, and 73.08%, the accuracy deficiency for the averaging method is 1.13% and 2.67%, and the accuracy deficiency for the maximum value method is 1.34% and 3.16%, respectively. Interestingly, on average, more features result in lower accuracy deficiency, in which the deficiency in accuracy is proportional to the size of the feature. The main advantage of Compact-Optimum-1 is that only the feature size is half that of Compact-Optimum-2, providing a low resource requirement with an estimated two-times-faster computing time.

The compact feature strategy experiment results indicate that the maximum selection or averaging of the feature vector could minimize redundant information in a feature vector; as a result, the proposed compact feature strategy achieves an acceptable trade-off condition in terms of reducing feature size and maintaining accuracy. Furthermore, the experiment results in Table 5 also provide an alternative strategy to determine the best accuracy between the maximum selection or averaging method for each feature extraction, which consists of the maximum value for the HOG feature and averaging for others, named the Combine-AMV method.

At this stage, the compact feature from each handcrafted-feature extraction is in a fit state for feature fusion. Although Compact-Optimum-2 has a higher level of accuracy and lower accuracy degradation than Compact-Optimum-1, Compact-Optimum-1 requires fewer resources and has a faster computation time. Both are relevant to the final objective of the proposed method: to achieve high accuracy with minimum feature size. Therefore, the best feature group and method to use to extract compact features have yet to be determined, and this will be carried out in the fusion strategy experiment, which will include both Compact-Optimum-1 and Compact-Optimum-2 with three compact strategies: the maximum value, averaging, and Combine-AMV methods.

### 4.2.3. Fusion-at-the-Feature-Level Experiment

The compact feature strategy produces two compact-optimum features in an appropriate state for the feature fusion feature strategy. This experiment evaluates and analyzes fusion feature schemes based on Compact-Optimum-1 and Compact-Optimum-2, focused on the trade-off between accuracy and feature size. The schema-1 fusion combines the compact feature configuration, which has a minimum length (Compact-Optimum1), resulting in a feature length of 4790, which is a combination of MLBP [50 × 50], HOG [40 × 40], Color HIST [50 × 50], and SURF [100 × 100]. Meanwhile, the schema-2 fusion combines the compact feature configuration, which has a maximum length (Compact-Optimum-2), which results in a feature length of 9455, which is a combination of MLBP [40 × 40], HOG [25 × 25], Color HIST [40 × 40], and SURF [50 × 50]. The compact-optimum feature from the maximum value, averaging, and combined methods are used for the fusion strategy experiment, and there are no significant performance differences from the previous experiment. To address the contradiction of feature size and accuracy decrement, the fusion from Optimum-1 and Optimum-2 serves as the baseline for Compact-Optimum-1 and Compact-Optimum-1, respectively.

From the experiments, as seen in Table 6, on average, the Schema-1 and Schema-2 groups achieve accuracy levels of 81.00% and 81.35%, with a decrement of 0.28% and 0.56%, respectively. This result contrasts with the compact feature strategy experiment, in which the configuration with more features experiences a less significant decrease in performance. Although, on average, a higher level of accuracy is achieved by Schema-2, the differences in accuracy are not significant, only being 0.35%. However, the feature size is as much as 1.97 times higher than Schema-1, which causes a higher demand for computational

resources and suggests that the Schema-2 processing time is twice that of Schema-1. The objective of the proposed method is to produce compact features with high accuracy and minimum feature size; therefore, considering the trade-off between accuracy and feature size, the Schema-1 group is selected as the having best configuration. Meanwhile, based on Table 6, the compact feature strategy using the averaging method achieves a higher level of accuracy in Schema-1 and Schema-2. This indicates that averaging the feature vector from several variations in a spatial location is more effective in reducing the redundant information while maintaining discriminant information compared with the maximum value, which assumes the peak of a signal always represents high discriminant information. Furthermore, the Combine-AMV method, expected to provide the best results, achieves slightly lower accuracy than the applied averaging method on all features in Schema-1 but achieves the best accuracy in Schema-2. Therefore, further investigation needs to be carried out, especially with regard to the compact feature strategy method, which could provide a high level of accuracy with a minimum feature size.

**Table 6.** The experiment results from the fusion strategy.

| Fusion Schema | Fusion From | Compact Feature Strategy | Feature Size | Ethnicity Accuracy | Delta Acc. |
|---|---|---|---|---|---|
| Schema-1 | Compact-Optimum-1 | Averaging | 4790 | 81.12% [1] | 0.16% [4–1] |
| Schema-1 | Compact-Optimum-1 | Maximum Value | 4790 | 80.89% [2] | 0.39% [4–2] |
| Schema-1 | Compact-Optimum-1 | Combine-AMV | 4790 | 80.98% [3] | 0.30% [4–3] |
| | Fusion-Optimum-1 | | 13,976 | Avg. 81.00% 81.28% [4] | Avg. 0.28% - |
| Schema-2 | Compact-Optimum-2 | Averaging | 9455 | 81.69% [5] | 0.22% [8–4] |
| Schema-2 | Compact-Optimum-2 | Maximum Value | 9455 | 80.50% [6] | 1.41% [8–5] |
| Schema-2 | Compact-Optimum-2 | Combine-AMV | 9455 | 81.86% [7] | 0.05% [8–7] |
| | Fusion-Optimum-2 | | 29,720 | Avg.81.35% 81.91% [8] | Avg. 0.56% - |

[1,2,3,4,5,6,7,8] address specific values in the table.

Based on the experiment results regarding feature fusion and referring to the objective of the proposed method (to produce compact features with high accuracy levels and minimum feature sizes), the configuration using Fusion-Compact-Optimum-1 with the averaging method, named Compact-Fusion AVG, is selected as an optimum solution for the ethnicity classification task.

4.2.4. Experiment on Testing Set

The evaluation of the best feature configuration (Compact-Fusion AVG) on the testing set is the final stage in the initial experiment. From the previous experiment, the best feature configuration to extract compact features is the averaging value method. As a reminder, the UTKFace Preprocessing dataset used in this study consists of 22,332 data, which are split into a training set (11,183), validation set (5572), and test set (5577). The previous experiment on the training and validation sets aimed to find the optimum configuration by adjusting parameters and configuration. Therefore, the evaluation of the testing set aims to determine the generalization capability of the learning model and configuration parameters from previous experiments. Finally, the testing set results are compared with the best independent handcrafted features (single feature) and fusion from the optimum feature to show a broad overview of the proposed framework.

The experiment results in Table 7, which compare the proposed method, the best of independent handcrafted features, and fusion from the optimum feature, show that the SVM learning model produces a similar accuracy rate on the testing and validation sets for all feature representations. Furthermore, the experiment results in Table 7 show the accuracy level from Compact-Fusion AVG with 4790 features (6.20 and 2.98 smaller than Fusion-Optimum-1 and Fusion-Optimum-2) is included in the top three, with slightly lower accuracy of about 0.29% and 1.12%. Generally, the proposed compact-fusion feature achieves higher accuracy than independent handcrafted features with smaller feature sizes. This result shows that the proposed framework has a sufficient generalization capability to handle unseen data. It offers a promising performance with low-cost resource requirements.

**Table 7.** Comparison of the proposed framework with handcrafted features and fusion from optimum.

| No. | Feature Representation | Feature Size | Validation Acc. | Testing Acc. |
|---|---|---|---|---|
| 1 | Compact-Fusion AVG | 4790 | 81.12% | 82.03% |
| 2 | MLBP [25 × 25] | 15,104 | 80.20% | 80.96% |
| 3 | HOG [25 × 25] | 13,824 | 77.39% | 78.79% |
| 4 | Color HIST [25 × 25] | 15,104 | 77.51% | 78.36% |
| 5 | SURF-based [50 × 50] | 4049 | 78.32% | 79.20% |
| 6 | Fusion-Optimum-1 | 13,976 | 81.28% | 82.32% |
| 7 | Fusion-Optimum-2 | 29,720 | 81.91% | 83.15% |

In an ethnicity classification task, it is essential to examine the recognition performance for each ethnicity to determine whether the proposed classification method contains bias for a specific ethnicity. Therefore, precision, recall, and F1-score are calculated for every ethnicity class, as seen in Table 8 and the confusion matrix reported in Figure 12. The experiment result in Table 8 shows that the proposed method has a fair accuracy value for each class with an average F1-Score of 73.32%. However, the proposed method performs poorly for the Others class; the main reason for the deficiency is unbalanced data and ambiguous labels. The ambiguous labels, as mentioned in the dataset description, mean that the Others class contains several ethnicities: Hispanic, Latino, and Middle Eastern. From the confusion matrix in Figure 12, the ethnic group pairs with the most significant misclassification for each class are: White–Indian, Black–White, Black–Indian, Asian–White, Indian–White, and Others–White. This result confirms our earlier assumptions regarding the challenges in ethnicity classification, where the feature representation must handle the appearance variance in the UTKFace dataset. Nevertheless, the multi-feature in this paper that compromises pixel color, texture, and pixel pattern has yet to be able to reduce misclassification due to ambiguous labels and insufficient sample data.

**Table 8.** Precision-recall and F-1 score for compact-fusion AVG.

| No. | Ethnicity | Number of Data | Precision | Recall | F1-Score |
|---|---|---|---|---|---|
| 1 | White | 2.414 | 83.93% | 90.43% | 87.06% |
| 2 | Black | 1.028 | 84.77% | 87.74% | 86.23% |
| 3 | Asian | 774 | 86.76% | 83.85% | 85.28% |
| 4 | Indian | 952 | 75.65% | 78.99% | 77.29% |
| 5 | Others | 409 | 52.35% | 21.76% | 30.74% |
| | Average | | 76.69% | 72.55% | 73.32% |

### 4.3. The Ablation Study

This study focuses on feature representation using multi-handcrafted features: MLBP, HOG, Color HIST, and SURF-based features. The ablation study to investigate the contribution of each handcrafted feature to accuracy in ethnicity classification is conducted on Compact-Fusion AVG. The ablation study alternately excludes one handcrafted feature from the feature representation, resulting in four feature-representation variations, named:

Compact-Fusion-AVG minus MLPB, Compact-Fusion-AVG minus HOG, Compact-Fusion-AVG minus Color, and Compact-Fusion-AVG minus SURF-based. A feature's contribution level is measured as the difference between accuracy with a full feature and a minus-one feature. The result from the ablation experiment is shown in Figure 13.

## Predicted Class

|  | White | Black | Asian | Indian | Others |
|---|---|---|---|---|---|
| White | 2183 | 58 | 45 | 98 | 30 |
| Black | 51 | 902 | 14 | 48 | 13 |
| Asian | 61 | 29 | 649 | 22 | 13 |
| Indian | 125 | 34 | 16 | 752 | 25 |
| Others | 181 | 41 | 24 | 74 | 89 |

Actual Class

**Figure 12.** Confusion matrix for compact-fusion AVG.

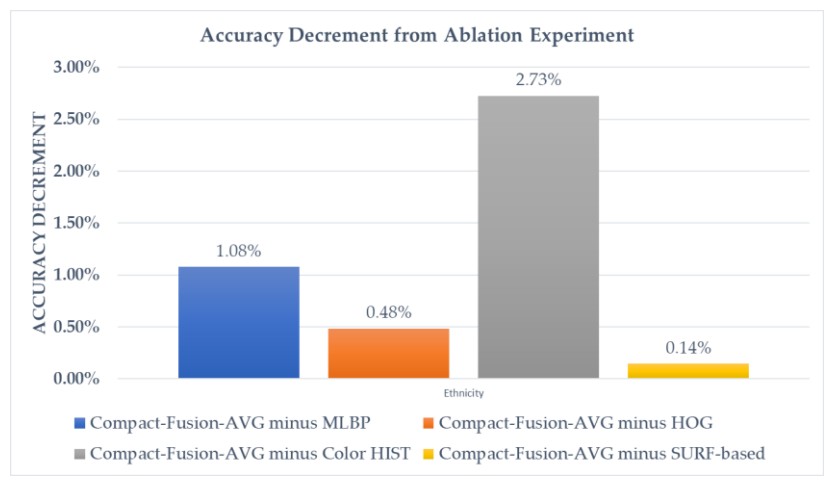

**Figure 13.** Accuracy decrease from the ablation experiment.

Based on Figure 13, the SURF-based features provide the lowest contribution, with only 256 features, compared to the MLBP, HOG, and Color HIST features. Moreover, the experiment results indicate that Color HIST features are most important in ethnicity classification. When Color HIST features are not included in the feature representation, there is a relatively high decrease in accuracy compared to other features, being as much as 2.73%. This experiment result is consistent with the theory that skin color is one of the phenotypic features that can be used to classify ethnicity.

### 4.4. Comparison with Feature-Reduction Method

The effectiveness of the strategy for compact feature representation was compared with three proven conventional feature-reduction algorithms, which are NCA (neighborhood component analysis), RICA (reconstruction-independent component analysis), and PCA (principal component analysis). NCA, PCA, and RICA were used to reduce Fusion-Optimum-1 features, which included MLBP [50 × 50], HOG [40 × 40], Color HIST [50 × 50], and SURF [100 × 100]. As a result, the size of the Fusion-Optimum-1 feature was reduced

from 13,976 to 4780 via NCA, PCA, and RICA. Figure 14 shows the experimental results comparing the Compact-Fusion AVG with the Fusion-Optimum-1+NCA, Fusion-Optimum-1+PCA, and Fusion-Optimum-1+RICA. The experiment results show that Compact-Fusion AVG outperforms other feature-reduction methods with slightly higher accuracy than PCA; this indicates that the compact-fusion feature has comparable discriminant information to the reduced feature result from PCA. In addition, the compact feature from the proposed method has trace-back characteristics that maintain the spatial location and provide a better understanding of the data generation process through the single data process.

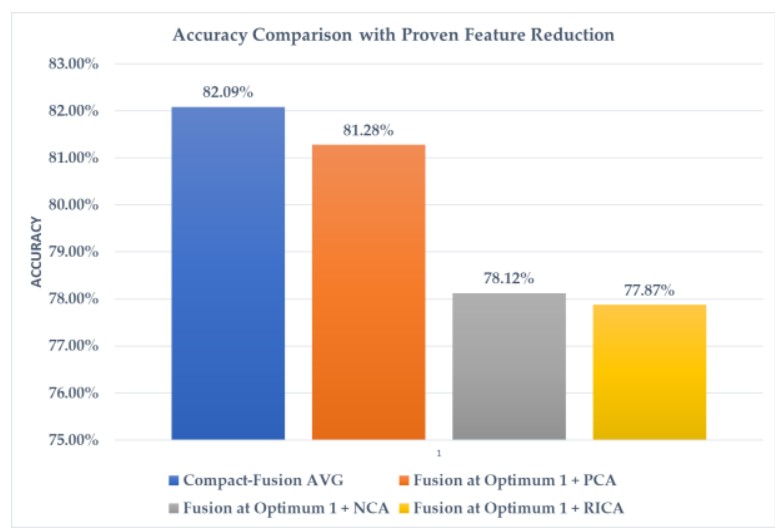

**Figure 14.** Comparison of compact-fusion AVG with other feature-reduction methods.

*4.5. Cross-Dataset Experiment*

For the cross-dataset experiment, four models are trained and tested pairwise. Model_1 is trained and tested on UTKFace Preprocessing-A, Model_2 is trained and tested on Fair Face Subset-A, Model_3 is trained on UTKFace Preprocessing-A and tested on Fair Face Subset-A, and Model_4 is trained on Fair Face Subset-A and tested on UTKFace Preprocessing-A. The results of the cross-dataset experiment can be seen in Table 9.

**Table 9.** The results of the cross-dataset experiment.

| | | **Trained On** | | | |
| --- | --- | --- | --- | --- | --- |
| | | UTKFace Preprocessing-A | | Fair Face Subset-A | |
| | | Model_1 | | Model_4 | |
| Tested on | UTKFace Preprocessing-A | Accuracy 89.14% | F1-Score 88.11% | Accuracy 75.95% | F1-Score 73.01% |
| | | Model_3 | | Model_2 | |
| | Fair Face Subset-A | Accuracy 55.31% | F1-Score 53.33% | Accuracy 73.87% | F1-Score 73.53% |
| | Average | 72.23%, | | 74.91% | |

Table 9 shows that the model that is trained with the same dataset as the testing set achieves the best accuracy. For example, in Model_1 (trained and tested on UTKFace Preprocessing-A), the highest accuracy is 89.14%, far surpassing Model_4 (trained on Fair Face Subset-A and tested on UTKFace Preprocessing-A) at 75.56%. Meanwhile, the highest accuracy in Model_2 (trained and tested on Fair Face Subset-A) is 73.87%, far surpassing Model_3 (trained on UTKFace Preprocessing-A and tested on Fair Face Subset-A) at 55.31%. Moreover, the cross-dataset experiment shows that the accuracy from the

model trained with UTKFace Preprocessing-A performs poorly when tested on different datasets compared with the model trained with Fair Face Subset-A. Finally, the dataset generalization is evaluated from the average accuracy of the models trained with it. As seen in Table 9, the average accuracy level from two models trained on UTKFace Preprocessing-A and Fair Face Subset-A is 72.23% and 74.91, respectively. These results indicate that the Fair Face Subset-A has better generalization than UTKFace Preprocessing-A when used in a cross-dataset. However, further investigation needs to be carried out to measure the generalization of the dataset used for cross-dataset evaluation.

The proposed compact feature framework shows suitable generalization, which can be seen from the accuracy of Model_2, which uses the compact feature configuration obtained from UTKFace and achieves an accuracy level of 73.87%, slightly lower than the reported accuracy of 75.40% by Karkkainen and Joo in [3], which was evaluated in the entire set. This condition probably occurs because we limit the number of data used, as with UTKFace, which means the data sample is insufficient to build a model to overcome the variation in the Fair Face dataset.

### 4.6. Classifier Comparison

The initial classifier used in the experiment is the SVM multi-class and implemented OVA schemes (One Versus All). The SVM learning parameter used is a polynomial kernel with order 3, C = 1, and a kernel scale set as auto (which uses a heuristic procedure to select the scale value with a random number set to 5). For further analysis, the comparison experiment is conducted with four types of classifiers: support vector machine (SVM), linear discriminant analysis (LDA), random forest (RF), and ensemble tree with Total-Boost to provide a comprehensive analysis.

The kernel parameters tested for SVM are: linear, polynomial order = 3, and RBF with a kernel scale of 69.21 (square root from feature dimension, which is 4790) and C = 1. For LDA, the kernels tested are linear and pseudolinear. Random forest is evaluated using a grid search with the number of trees in {100, 300, 750, 1000} and the number of predictors in {690, 863, 1035, 1208, 1380}. Finally, the grid search with the number of learners in {30, 50, 100, 300} and max split in {20, 50, 100, 200, 500} are tested for ensemble tree with Total-Boost.

Based on the experimental results shown in Table 10, the SVM-OVA-based classifier with the parameter setting of the polynomial kernel with the order of 3, C = 1, and a kernel scale of 69.21 (square root from feature dimension) achieves the best accuracy level of 82.19%. Meanwhile, the LDA classifier with a linear kernel obtains an accuracy level of 80.11%. Finally, for the RF and ensemble tree classifier, the accuracy obtained from the grid-search configuration is around 70%, estimated to be due to the large size of the dataset and the fact that the parameter tuning is not optimal yet, which requires further investigation.

**Table 10.** Experiment result for classifier comparison.

| Classifier Tested | Accuracy |
| --- | --- |
| SVM-OVO, linear kernel | 78.82% |
| SVM-OVO, polynomial kernel order 3 | 81.82% |
| SVM-OVO, RBF kernel | 79.74% |
| SVM-OVA, linear kernel | 79.68% |
| SVM-OVA, polynomial kernel order 3 | 82.19% |
| SVM-OVA, RBF kernel | 81.05% |
| LDA with linear kernel | 80.11% |
| LDA with linear kernel | 71.63% |
| RF: Grid search, best at nTree = 100, the number of predictors = 1035 | 70.29% |
| Ensemble Tree, Total-Boost, n Learner= 100, maxSplit = 200 | 71.01% |

### 4.7. Comparison with SOTA

With the development of hardware resources and the emergence of large datasets providing ethnicity labels, neural networks, especially CNNs, are more often used than the

handcrafted approach to developing ethnicity classification solutions. However, fair and exact comparisons with other research are hard to achieve as different authors use different validation protocols, feature-extraction methods, and learning models. The relative position of the proposed method with the results of other studies that used the UTKFace dataset for ethnicity classification tasks is reported in Table 11. The proposed compact-fusion AVG method achieves comparable accuracy in multi-class ethic classification with an accuracy level above 80%, and it represents a promising alternative solution with the requirement of low-cost resources compared with deep-learning approaches.

**Table 11.** Comparative result accuracy of state of the art for ethnicity classification on the UTKFace for five classes.

| Paper | Ethnicity | Method | Feature Size | Accuracy |
|---|---|---|---|---|
| Proposed method | White, Black, Asian, Indian, and Others<br>White, Black, Asian, and Indian | Compact-Fusion + SVM | 4790 | 82.19%<br>89.14% |
| Belcar et al. (2022) [2] | White, Black, Asian, Indian, and Others | CNN Based | 11,520 * | 80.34% ** |
| Ahmed et al. (2022) [4] | Caucasian, African, Asian, Indian | R-Net | 3136 * | 77.5% *** |
| Hamdi and Moussaoui (2020) [5] | White, Black, Asian, Indian, and Others | CNN Based | 8192 * | 78.88% |
| Al-Azani and El-Alfy (2019) [6] | Asian, Indian, and Others | HOG | 5292 * | 69.68% |

* Estimated based on information in the paper; for deep-learning approach, it is the output of the last max pool layer. ** Only used the middle part of the face. *** Average accuracy from age group experiment.

## 5. Conclusions

The proposed compact-fusion feature framework shows the capability of producing compact features that achieve optimum performance with minimum feature size and competitive accuracy. In the initial experiment, the proposed method achieved significant improvement in accuracy, from 80.96% to 82.03% (+1.07%), and a 68.29% (from 15,104 to 4790) reduction ratio compared with the best accuracy for single handcrafted features (MLBP [25 × 25]). Furthermore, the proposed feature reduction has comparable discriminant information with PCA, which means that the proposed feature reduction successfully removes redundant information. Finally, combined with the SVM-OVA-based classifier with the parameter setting of the polynomial kernel with the order of 3, C = 1, and kernel scale of 69.21 (square root from feature dimension), and with a feature size of 4790, the proposed method with the SVM-OVA classifier achieves accuracy levels of 89.14% and 82.19%, respectively, in the UTKFace dataset with four and five classes, and the Fair Face dataset with four classes. This result is comparable with the method based on the deep-learning approach.

The proposed compact-fusion feature framework is a tailor-made design, using the feature-extraction method, compact feature strategy, and feature fusion. The simple process of producing compact features with averaging and a grid-based process gives trace-back characteristics and provides a better understanding of the data generation. The experiment results indicate that the conventional approach using handcrafted features is still promising for use in ethnicity classification tasks. Furthermore, with a proper analysis and strategy, the solution based on handcrafted features can achieve suitable performance with a low-cost demand for computing resources. However, the limitation of the proposed framework is the absence of a parameter to control the level of the reduction ratio for flexibility in application.

In future work, to achieve a higher reduction ratio, this method can be preceded by using the feature-selection/feature-reduction method, including importance measurement, RICA, NCA, and PCA, to produce more compact features for ethnicity classification. Furthermore, the exploitation of a grid-based approach allows grids with low discriminant information to be discarded, which results in a higher reduction ratio. Meanwhile, to achieve a higher level of accuracy, it can be combined with a deep-learning approach while maintaining the explainable characteristics of the handcrafted feature.

**Author Contributions:** Conceptualization, T.A.B.W., R.M. and A.I.K.; methodology, T.A.B.W., R.M. and A.I.K.; software, T.A.B.W.; validation, T.A.B.W., R.M. and A.I.K.; formal analysis, T.A.B.W.; investigation, T.A.B.W.; resources, T.A.B.W.; data curation, T.A.B.W.; writing—original draft preparation, T.A.B.W.; writing—review and editing, T.A.B.W., R.M. and A.I.K.; visualization, T.A.B.W.; supervision, R.M. and A.I.K.; project administration, R.M. and A.I.K.; funding acquisition, R.M. All authors have read and agreed to the published version of the manuscript.

**Funding:** This research was funded by ITB-Indonesia through Penelitian, Pengabdian kepada Masyarakat dan Inovasi ITB (P2MI-ITB) grant.

**Institutional Review Board Statement:** Not applicable.

**Informed Consent Statement:** Written informed consent was obtained from the study participants for the publication of their details.

**Data Availability Statement:** Not applicable.

**Acknowledgments:** Not applicable.

**Conflicts of Interest:** The authors declare no conflict of interest.

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
