# Peer review of "Compact-Fusion Feature Framework for Ethnicity Classification"

_informatics, doi:10.3390/informatics10020051_

Round 1
Reviewer 1 Report (Previous Reviewer 1)
I have thoroughly examined the revised version of the manuscript and the responses to the review.
To write a better paper, it appears that additional work is needed.
I would like to add a few additional comments.
1. The abstract section needs to be revised to be more concise and easier to understand. In particular, it would be advisable to rewrite the sentences from line 14 to line 21.
2. The introduction section should eliminate irrelevant or redundant content and provide a concise and clear summary of the problem addressed in this study. Additionally, the authors' claimed contributions are not easily discernible. I recommend rewriting the introduction to make it concise and clear.
3. In the related work section, authors should concisely describe previous research that inspired or directly related to this research. Do all the cited relevant studies have a direct connection to this study or are they necessary for comparison? I'm wondering if listing too many prior studies is merely a matter of enumeration.
4. It would be advisable to merge Sections 3 and 4 into one section. The content of Section 3 should be appropriately relocated within the framework description in Section 4. Additionally, it is recommended to cite and remove any duplicated or unnecessary content.
5. It would be advisable to move the dataset description in Section 3.5 to the experimental section.
Extensive editing of English language required.
Please pay more attention to the use of punctuation and similar elements.
I recommend getting proofreading from a native English speaker.
Author Response
Please see the attachment

Reviewer 2 Report (Previous Reviewer 2)
1) As it was the revised manuscript so I would say
Perform Ablation study as it is still missing
2) You can cite the following paper as it is relevant
Ahmed Sheikh Bilal, Syed Farooq Ali, Jameel Ahmad, Muhammad Adnan, and Muhammad Moazam Fraz. "On the frontiers of pose invariant face recognition: a review." Artificial Intelligence Review, Journal (2019), vol. 53: 1-64. [Online]. Available: https://doi.org/10.1007/s10462-019-09742-3
Round 2
Reviewer 1 Report (Previous Reviewer 1)
Dear Authors,
I accept the response to my review comments.
Please review the English sentences again carefully and make sure there are no errors in the last revision
This manuscript is a resubmission of an earlier submission. The following is a list of the peer review reports and author responses from that submission.
Round 1
Reviewer 1 Report
The authors proposed a compact-fusion feature framework for ethnic classification.
In this paper, they presented a hand-craft feature-based approach with lower computational cost compared to deep learning-based methods.
However, the legitimacy of the research purpose claimed in this paper is weak, and it is difficult to find theoretical and practical important contributions.
1. In the introduction section, it is necessary to clearly state the definition of the problem to be solved through this study and the theoretical or practical contribution of the results of this study.
2. In the related work section, authors should concisely describe previous research that inspired or directly related to this research. It is necessary to revise the related works sections, including the studies to which this study is compared.
3. In proposed framework section, the methodology proposed in this paper is described at a too abstract level, and the proposed method is not clearly described.
4. What is "predetermined channel selection and extraction configuration" on page 5, line 185 of the manuscript?
5. What is "predetermined feature extraction configuration" on page 5, line 193 of the manuscript?
6. The authors mentioned that "The compact feature strategy stages produce features that minimize redundant information from an optimum feature". Is the optimum feature mentioned above related to equation (1) and equation (2)? It's difficult to understand how redundant information can be eliminated just by explaining the formula.
7. Regarding Figure 2 and Algorithm 1, the exact definition and extraction method of Optimum features should be described. In addition, the exact definition and extraction method of compact features should be explained. Finally, a detailed description of the fusion features is required.
8. What is the preprocessing mentioned in section 4.2? Why did the number of datasets decrease from 23,750 to 22,323?
9. It is difficult to determine which claims are supported by the experimental results presented by the authors.
Reviewer 2 Report
1) This paper used only one dataset. It might be a good idea if he used multiple datasets.
2) Figure 6 looks very blurry. It should be drawn again.
3) Compare your proposed approach with deep learning algorithms.
4) Why you have not proposed some deep learning algorithm?
5) Also perform the ablation study of your proposed approach
6) Why you used SVM classifier? Generate the results by replacing SVM classifier by Adaboost, ID3, Decision Table, J48, Bagging, Boosting, and Blending. After generating the results, compare it with SVM.